# Parallel adaptation in autopolyploid *Arabidopsis arenosa* is dominated by repeated recruitment of shared alleles

Veronika Konečná [1,2], Sian Bray[3], Jakub Vlček [1,4,5], Magdalena Bohutínská[1,2], Doubravka Požárová[1], Rimjhim Roy Choudhury[6,7], Anita Bollmann-Giolai [8], Paulina Flis [3], David E. Salt [3], Christian Parisod [6], Levi Yant [9,10 ✉] & Filip Kolář [1,2,10 ✉]

Relative contributions of pre-existing vs de novo genomic variation to adaptation are poorly understood, especially in polyploid organisms. We assess this in high resolution using autotetraploid *Arabidopsis arenosa*, which repeatedly adapted to toxic serpentine soils that exhibit skewed elemental profiles. Leveraging a fivefold replicated serpentine invasion, we assess selection on SNPs and structural variants (TEs) in 78 resequenced individuals and discover significant parallelism in candidate genes involved in ion homeostasis. We further model parallel selection and infer repeated sweeps on a shared pool of variants in nearly all these loci, supporting theoretical expectations. A single striking exception is represented by *TWO PORE CHANNEL 1*, which exhibits convergent evolution from independent de novo mutations at an identical, otherwise conserved site at the calcium channel selectivity gate. Taken together, this suggests that polyploid populations can rapidly adapt to environmental extremes, calling on both pre-existing variation and novel polymorphisms.

[1] Department of Botany, Faculty of Science, Charles University, Prague, Czech Republic. [2] The Czech Academy of Sciences, Institute of Botany, Průhonice, Czech Republic. [3] Future Food Beacon and School of Biosciences, University of Nottingham, Nottingham, UK. [4] Biology Centre, Czech Academy of Sciences, České Budějovice, Czech Republic. [5] Department of Zoology, Faculty of Science, University of South Bohemia, České Budějovice, Czech Republic. [6] Institute of Plant Sciences, University of Berne, Bern, Switzerland. [7] Department of Systematic and Evolutionary Botany, University of Zurich, Zurich, Switzerland. [8] John Innes Centre (JIC), Norwich Research Park, Norwich, UK. [9] Future Food Beacon and School of Life Sciences, University of Nottingham, Nottingham, UK. [10] These authors jointly supervised this work: Levi Yant, Filip Kolář. ✉email: levi.yant@gmail.com; filip.kolar@natur.cuni.cz

Rapid adaptation to novel environments is thought to be enhanced by the availability of genetic variation; however, the relative contribution of standing variation versus the role of novel mutation is a matter of debate[1,2], especially in higher ploidy organisms. Whole-genome duplication (WGD; leading to polyploidisation) is a major force underlying diversification across eukaryotic kingdoms, seen most clearly in plants[3–5] with various effects on genetic variation[6–8]. While WGD is clearly associated with environmental change or stress[5,9], the precise impact of WGD on adaptability is largely unknown in multicellular organisms, and there is virtually no work assessing the evolutionary sources of adaptive genetic variation in young polyploids. Work in autopolyploids, which clearly isolate effects of WGD from hybridisation (which is confounded in allopolyploids), indicates that subtle genomic changes may follow WGD alone[8,10,11], which raises the question of when their adaptive value may originate.

Autopolyploidy is expected to alter selective and adaptive process in many ways, but a dearth of empirical data prevents synthetic evaluation. Besides immediate phenotypic[12–14] and genomic[10,11] changes following WGD, theory is unsettled regarding how adaptation proceeds as the autopolyploid lineage diversifies and adapts to novel challenges. On the one hand, autopolyploids can mask deleterious alleles and accumulate cryptic allelic diversity[7]. In addition, the number of mutational targets is multiplied in autopolyploids, meaning that new alleles are introduced more quickly[6,15,16]. This could promote adaptation[6,17]. On the other hand, reduced rates of allele frequency changes may retard adaptation[6,18], particularly for de novo mutations, which emerge in a population at initially low frequencies[19]. Recent advances in theory and simulations suggest potential solutions to this controversy. Polyploidy may promote adaptation under scenarios of rapid environmental change (e.g. colonisation of challenging habitats) when selection is strong and originally neutral or mildly deleterious alleles standing in polyploid populations may become beneficial[5,20]. However, empirical evidence supporting this scenario is fragmentary. There is broad correlative evidence that polyploids are good colonisers of areas experiencing environmental flux (e.g. the Arctic[21,22], stressful habitats[8,23,24], and heterogeneous environments[25]). However, the genomic basis of such polyploid adaptability—and whether their primary source of adaptive alleles is high diversity (standing variation) or large mutational target size (de novo mutations)—remains unknown.

We focus on natural autotetraploid *Arabidopsis arenosa* populations repeatedly facing one of the greatest environmental challenges for plant life—naturally toxic serpentine soils. Serpentines occur as islands in the landscape with no intermediate habitats and are defined by peculiar elemental contents (highly skewed Ca/Mg ratio and elevated heavy metals such as Cr, Co, and Ni), that may be further combined with low nutrient availability and propensity for drought[26]. *Arabidopsis arenosa* is a well-characterised, natural diploid-autotetraploid species with large and genetically diverse outcrossing populations[27]. The widespread autotetraploids, which originated from a single diploid lineage ~19–31k generations ago[8], harbour increased adaptive diversity genome wide[8] and currently occupy a broader ecological niche than their diploid sisters[28], including serpentine outcrops[29], railway lines[30,31], and contaminated mine tailings[32,33]. This makes *A. arenosa* a promising model for empirical inquiries of adaptation in autopolyploids[8,27]. As a proof of concept, selective ion uptake phenotypes and a polygenic basis for serpentine adaptation have been suggested from a single *A. arenosa* serpentine population[29]. However, limited sampling left unknown whether the same genes are generally (re)used and what is the evolutionary source of the selected alleles, i.e. leaving unresolved the evolutionary dynamics and mechanism underlying these striking adaptations. We ask specifically: (1) Does gene-level parallelism in autotetraploid *A. arenosa* dominantly reflect repeated sampling from the large pool of shared variation that is expected to be maintained in autopolyploids? and (2) Is repeated adaptation from novel mutations feasible in autotetraploid populations?

In this work, we deconstruct the sources of parallel adaptive variation in *A. arenosa*. First, we sample five serpentine/non-serpentine population pairs of autotetraploid *A. arenosa* and demonstrate rapid parallel adaptation by combining demographic analysis and reciprocal transplant experiments. Taking advantage of the power of this fivefold replicated natural selection experiment, we identify candidate adaptive loci from population resequencing data and find significant parallelism underlying serpentine adaptation. We then model parallel selection using a designated framework and statistically infer the evolutionary sources of parallel adaptive variation for all candidate loci. In line with theory, we find that shared variation is the vastly prevalent source of parallel adaptive variants in serpentine *A. arenosa*. However, we also discover an exceptional locus exhibiting footprints of selection on alleles originating from two distinct de novo mutations. In line with the latter hypothesis, this demonstrates that the rapid selection of novel alleles is still feasible in autopolyploids, indicating broad evolutionary flexibility of lineages with doubled genomes.

## Results

**Parallel serpentine adaptation.** First, we inferred independent colonisation of each serpentine site by different local *A. arenosa* populations. To do this, we resequenced five pairs of geographically proximate serpentine (S) and non-serpentine (N) populations covering all known serpentine sites occupied by the species to date (8 individuals per population on average, mean sequencing depth 21×; Fig. 1a, Supplementary Fig. 1, Supplementary Data 1, and Supplementary Tables 1–3). Phylogenetic, ordination, and Bayesian analyses based on nearly neutral fourfold-degenerate (4dg) sites demonstrated overall grouping of populations by spatial proximity, not by substrate. In all but one case, the adjacent S and N populations occupied sister position in the population tree and belonged to the same Bayesian cluster; only the population S3 occupied somewhat isolated position yet still within the lineage of Eastern Alpine populations (Fig. 1b and Supplementary Fig. 1). We thus further tested the independent colonisation of each serpentine site by coalescent simulations. Consistently over all possible pairwise iterations of S–N population pairs ($n = 10$), the scenario of independent colonisation of each serpentine site was more likely than any scenario assuming sister position of two S populations (Fig. 1c). Note that subsequent gene flow between substrate types within each S–N population pair was unlikely as the assumption of migration within each population pair had not significantly improved the model fit (Supplementary Fig. 2 and Supplementary Data 2). Reflecting the independent origin of the five S populations, we analysed each serpentine colonisation event separately in the following analyses to take into account neutral population structure in the data, using the spatially closest N population as a contrast where needed. The very low differentiation between S and proximate N populations and consistently low population split times (Table 1) indicate very recent, postglacial serpentine invasions. There is no evidence of bottleneck associated with colonisation, as S and N populations exhibited similar nucleotide diversity and Tajima's *D* values (Table 1).

To assess whether the colonisation of serpentines was accompanied by substrate adaptation, we combined ionomics

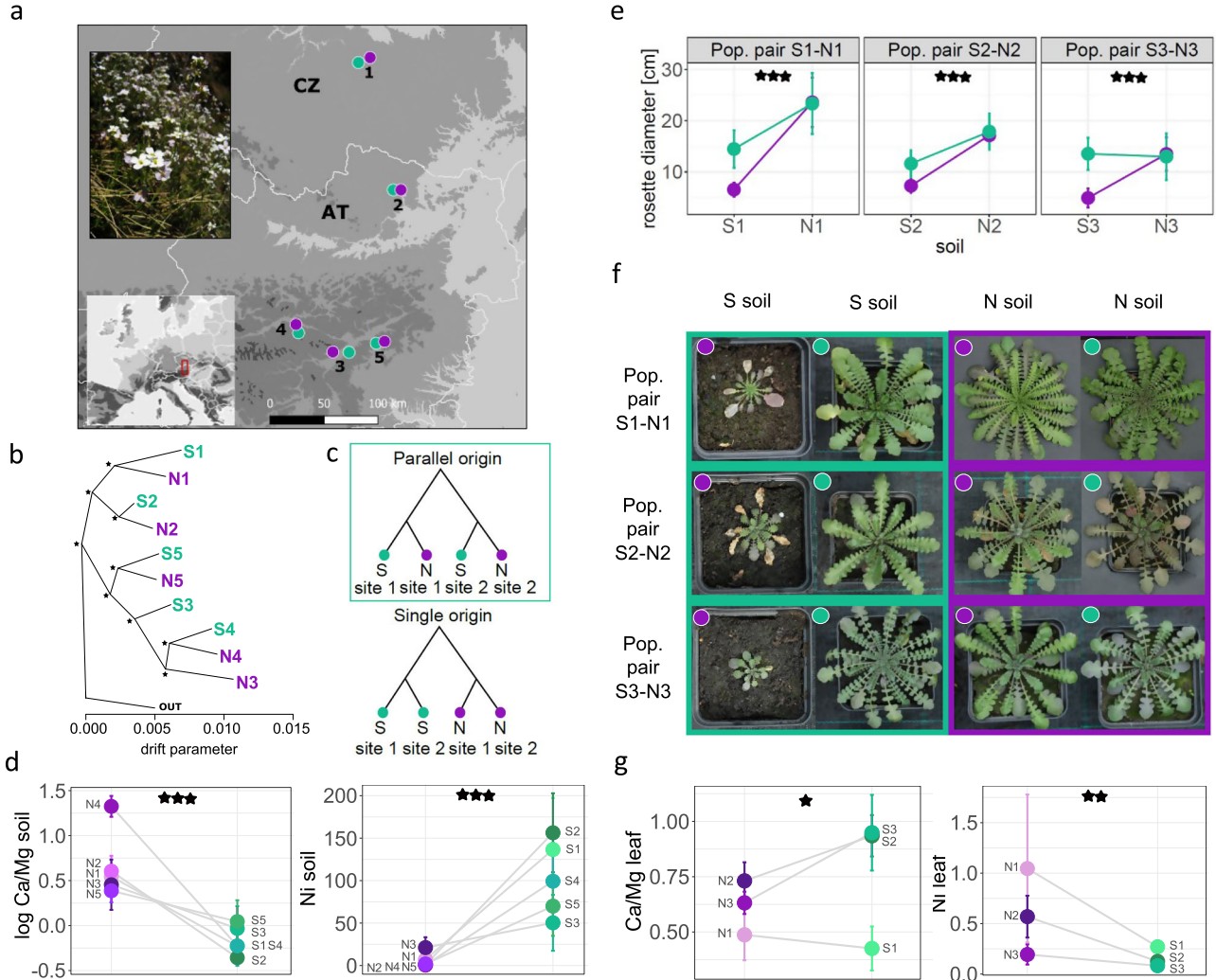

**Fig. 1 Parallel adaptation of *Arabidopsis arenosa* to challenging serpentine soils. a** Locations of the investigated serpentine (S, green) and non-serpentine (N, violet) populations sampled as spatially proximate pairs (numbers) in Central Europe with an illustrative photo of an S population (photo was taken by F. Kolář). **b** Allele frequency covariance graph of populations based on ~870,000 fourfold-degenerate SNPs; asterisks show the 100 bootstrap branch support. The outgroup (OUT) is represented by a tetraploid population from Western Carpathians, the ancestral area of tetraploid *A. arenosa*[93]. **c** Two contrasting evolutionary scenarios of serpentine colonisation compared in coalescent simulations; the topology assuming independent serpentine colonisations (framed in green) received the highest support consistently across all 10 pairwise combinations of S–N population pairs. **d** Differences in Ca/ Mg ratio and in Ni concentrations [μg/g] in S and N soils from the original sampling sites ($n = 78$ individual samples, one-way ANOVAs: $F_{1,77} = 26.5$, $p = 1.94e-06$ and $F_{1,77} = 117.4$, $p = 2.01e-16$ for Ca/Mg and Ni, respectively). **e** Differences in maximum rosette size of three population pairs attained after 3 months of cultivation in local serpentine and non-serpentine substrates (significance of the soil treatment × soil origin interaction in a two-way ANOVA is indicated: $F_{1,90} = 21.6$, $p = 1.17e-05$, $F_{1,96} = 12.3$, $p = 6.88e-04$, and $F_{1,85} = 42$, $p = 5.68e-09$ for population pairs 1, 2 and 3, respectively). **f** Example photos illustrating parallel growth response in the three population pairs to serpentine soils (green frame) depending on the soil of origin (dot colour) (photo was taken by V. Konečná). **g** Differences in ion uptake between originally S and N individuals when cultivated in serpentine soils; Ni concentrations were standardised by corresponding soil Ni values ($n = 28$ individual samples, one-way ANOVAs: $F_{1,27} = 6.2$, $p = 0.019$ and $F_{1,27} = 13.5$, $p = 0.001$ for Ca/Mg ratio and Ni, respectively). Points denote mean, error bars depict standard error of mean in charts **e**, **d**, **g**. Source data underlying Fig. 1d, g are provided as a Source data file.

with a reciprocal transplant experiment. First, using ionomic profiling of native soil associated with each sequenced individual, we characterised major chemical parameters differentiating on both substrates (Fig. 1d and Supplementary Figs. 3 and 4). Among the 20 elements investigated (Supplementary Fig. 3a), only the bioavailable concentration of Mg, Ni, Co, and Ca/Mg ratio consistently differentiated both soil types (Bonferroni-corrected one-way analysis of variance (ANOVA) taking population pair as a random variable). Serpentine sites were not macronutrient poor (Supplementary Table 4) and were not differentiated from non-serpentines by bioclimatic parameters (annual temperature, precipitation, and elevation; Supplementary

Fig. 3), indicating that skewed Ca/Mg ratios and elevated heavy metal content are likely the primary selective agents on the sampled serpentine sites[34–36].

We then tested for differential fitness response towards serpentine soil between populations of S versus N origin using reciprocal transplant experiments. We cultivated plants from three population pairs (S1–N1, S2–N2, and S3–N3) on both native soil types within each pair for three months (until attaining maximum rosette size), observing significantly better germination and growth of the S plants in their native serpentine substrate as compared to their closest N relatives. First, we found a significant interaction between soil type and soil of origin at germination

**Table 1 Between-population divergence and within-population diversity of the five investigated serpentine/non-serpentine population pairs inferred from genome-wide fourfold-degenerate single nucleotide polymorphisms.**

| Population pair | Divergence (generations)[a] | Pairwise $F_{ST}$ | Nucleotide diversity[b] | Tajima's $D$[b] |
|---|---|---|---|---|
| S1–N1 | (774) 4317 (6284) | 0.069 | 0.0292/0.0276 | 0.266/0.485 |
| S2–N2 | (500) 2690 (3826) | 0.029 | 0.0306/0.0285 | 0.131/0.372 |
| S3–N3 | (794) 3912 (6316) | 0.085 | 0.0307/0.0287 | −0.046/0.350 |
| S4–N4 | (812) 2918 (3542) | 0.057 | 0.0304/0.0297 | 0.132/0.177 |
| S5–N5 | (546) 3539 (4869) | 0.047 | 0.0292/0.0296 | 0.267/0.089 |

[a]Divergence between proximal S–N populations. Mean and 95% confidence intervals inferred by bootstrapping, estimated by coalescent simulations. Assuming 2-year generation time[93], all estimates indicate recent postglacial divergence.
[b]Genome-wide nucleotide diversity ($\pi$) and Tajima's D of each S–N population.

(generalised linear model (GLM) with binomial errors taking population pair as a random variable, $\chi^2 = 22.436$, $p < 0.001$), although the fitness disadvantage of N plants in serpentine soil varied across population pairs (Supplementary Fig. 5). During subsequent cultivation, we recorded zero mortality but found a significant interaction effect between soil treatment and soil of origin on growth, as approximated by maximum rosette sizes (two-way ANOVA taking population pair as a random variable, $F_{1,277} = 55.5$, $p < 0.001$, Fig. 1e; see Supplementary Fig. 6 for rosette size temporal development). Once again, the S plants consistently produced significantly larger rosettes (by 47% on average) than their N counterparts when grown in serpentine soil, indicating consistent substrate adaptation (Fig. 1e, f). Finally, we evaluated differences in Ni and Ca/Mg accumulation in leaves harvested on plants cultivated in serpentine soils. Consistent with adaptive responses to soil chemistry, we found higher Ca/Mg ratio and reduced uptake of Ni (lower leaf/soil ratio) in tissue of serpentine plants relative to their non-serpentine counterparts (Fig. 1g). Taken together, our demographic analysis complemented by transplant experiments support recent parallel serpentine adaptation of autotetraploid *A. arenosa* at five distinct sites, exhaustively covering all known serpentine populations of the species.

**Parallel genomic footprints of selection on serpentine at single-nucleotide polymorphisms (SNPs) and transposable elements (TE)s.** Using these five natural replicates of serpentine adaptation, we sought the genomic basis and evolutionary source of the parallel adaptations. To do this, we combined divergence scans and environmental association analysis to refine the list of loci for parallel selection modelling only to the candidates that repeatedly differentiated across multiple population pairs and were significantly associated with the selective soil environment. First, we identified initial inclusive lists of gene-coding loci exhibiting excessive differentiation between paired populations using 1% outlier $F_{ST}$ window-based scans (490–525 candidate genes per pair; details in 'Methods'; Supplementary Data 3). These most inclusive lists must be interpreted with caution, as they are based on a simple assumption that the most differentiated regions are under directional selection[37]. However, in support of their relevance a gene ontology (GO) analysis of the 2245 candidates from all five population pairs shows significant enrichment (Fisher's exact test; $p < 0.05$) of 'biological processes', 'molecular functions', and 'cellular components' considered relevant to serpentine adaptation[26,34], such as inorganic anion transport, ion homeostasis, post-embryonic development, and calcium transmembrane transporter activity (Fig. 2b and Supplementary Data 4).

To refine this broad list and pinpoint parallel evolution candidate genes, we overlapped these candidate gene lists across population pairs, identifying 207 'parallel differentiation candidates' that represent divergence outliers in at least two S–N population pairs. The level of parallelism was greater than

expected by chance for all pairs of S–N contrasts (Fisher's exact test; Fig. 2a and Supplementary Data 4) and we hereafter refer to this as 'significant parallelism'. Such a fraction of parallel gene candidates (0.02–0.04 out of all candidates from that particular population pair) is in line with other naturally adapting systems of comparable divergence[33,38–40]. The parallel differentiation candidates were significantly enriched ($p < 0.05$) for GO terms, such as regulation of ion transmembrane transport, voltage-gated calcium and potassium channel activity or plasma membrane (Supplementary Data 4). The absence of common candidates across all five population pairs may reflect a complex genetic basis of the traits allowing for the modulation of the same pathway by different genes in some populations. This is supported by significant functional parallelism, i.e. higher than random number of overlapping GO terms that were repeatedly identified by separate enrichment analyses of outlier gene list from each population pair (Supplementary Fig. 7 and Supplementary Data 4). Additionally, adaptation via partial (soft) sweeps, which are likely to occur in autotetraploids[19], might have further limited the power of our divergence scans in some loci and populations.

As a complementary approach, we inferred candidates directly associated with the distinctive chemical characteristics of serpentine soil by performing environmental association analysis using latent factor mixed models (LFMM)[41]. This analysis quantitatively determines the association between each soil elemental concentration and SNPs across the genome in both S and N populations at the level of individual plants (in total 78). We identified 2,809 genes (LFMM candidates) harbouring ≥1 SNP significantly associated with at least one distinctive serpentine soil parameter previously identified by ionomic analysis (Ca/Mg ratio, high Mg, Ni, and Co; Supplementary Data 5). Finally, we overlapped the LFMM candidates with the parallel differentiation candidates to produce a final refined list of 61 'serpentine adaptation candidates' (Fig. 2c, Supplementary Fig. 8, and Supplementary Data 6 and 7). This conservative approach aims to identify the strongest candidates underlying serpentine adaptation for further model-based inference of the sources of variation in the next section. We note that this approach discards population-specific (private) candidates and cases of distinct genetic architecture of a trait (e.g. distinct genes affecting the same pathway) and thus cannot quantify the overall genome proportion that evolves in parallel. Importantly, however, it also minimises false positives from population-specific selection and genetic drift.

These 61 serpentine adaptation candidates were significantly enriched ($p < 0.05$) for categories related, for example, to regulation of ion transmembrane transport, and specifically, voltage-gated calcium and potassium channel activity (Supplementary Data 7). Candidates included the *NRT2.1* and *NRT2.2* high-affinity nitrate transporters, which act as repressors of lateral root initiation[42,43]; *RHF1A*, which is involved in gametogenesis and transferase activities[44,45]; *TPC1*, a central calcium channel

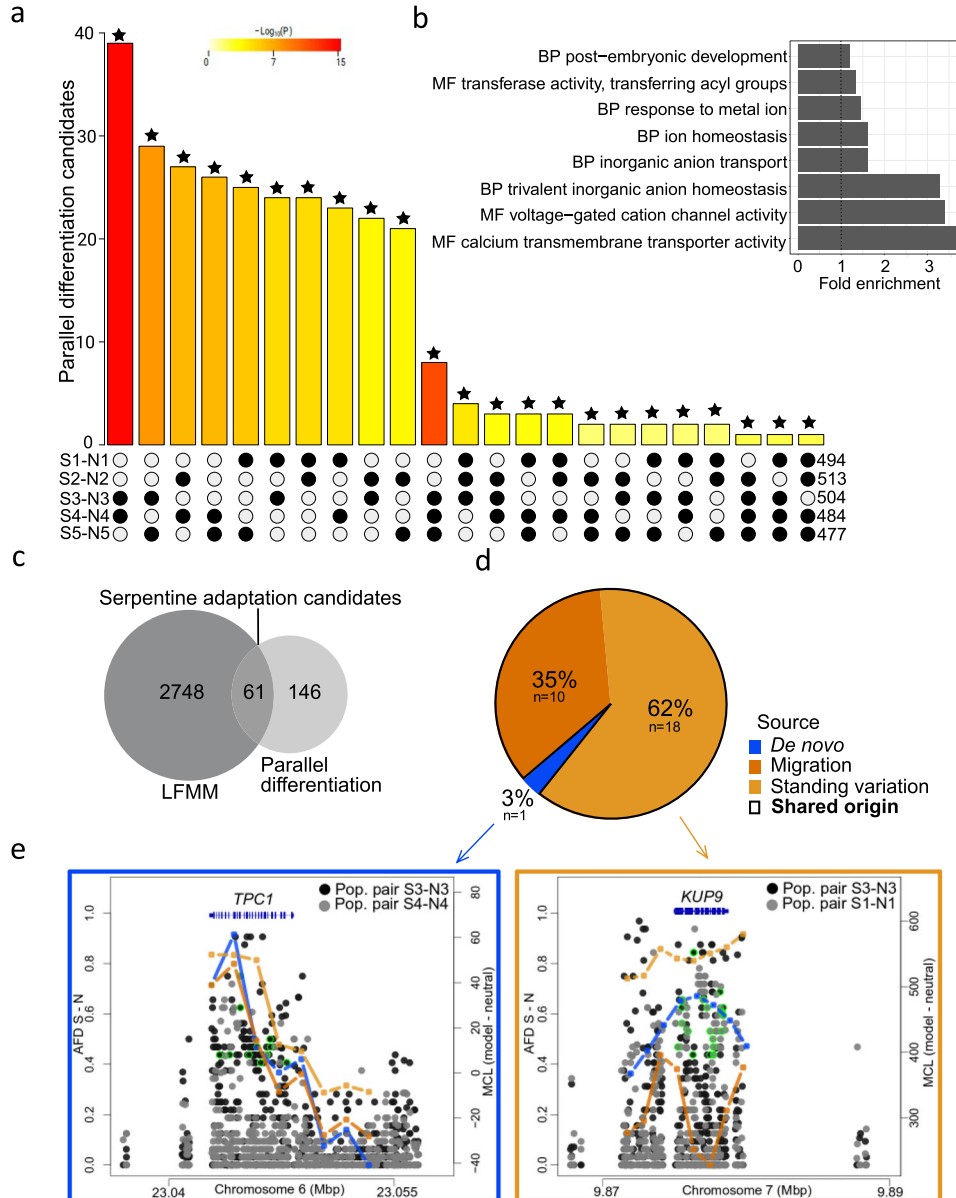

**Fig. 2 Parallel serpentine adaptation candidates and the sources of parallel variants in _A. arenosa_. a** Intersection of candidates from each population pair (S1–N1 to S5–N5) demonstrating more genes repeatedly found as candidates across two, three, and four population pairs than expected by chance alone (all intersections were significant at $p < 0.01$ (highlighted by asterisks), one-sided Fisher's exact test); note: the colour intensity of the bars represents the $p$ value significance of the intersections. **b** Gene ontology (GO) enrichment of the candidates (across all population pairs); GO categories: biological process (BP) and molecular function (MF); for complete list of GO terms, see Supplementary Data 4. **c** Overlap between parallel differentiation candidates and latent factor mixed model (LFMM) candidates resulting in 61 serpentine adaptation candidate genes. **d** Proportions of serpentine adaptation candidates originating from de novo mutations or being of shared origin out of the total of 29 cases of non-neutral parallelism as inferred by the Distinguishing among Modes of Convergence approach (DMC; see text for details). **e** Two examples of parallel candidate loci, illustrating nucleotide divergence and maximum composite log-likelihood (MCL) estimation of the source of the selected alleles in these particular loci inferred in DMC. Allele frequency difference (AFD) for locus with independent de novo mutations (left) and with parallel recruitment of shared ancestral standing variation (right). Left $y$-axis: AFD between S and N populations. Dots: AFD values of individual SNPs; bright green circles: non-synonymous SNPs with AFD $\geq 0.4$; lines (right $y$-axis): MCL difference between neutral versus parallel selection scenario following colour scheme in **d**; gene models are in blue.

that mediates plant-wide stress signalling and tolerance[46]; and potassium transporters _AKT5_ and _KUP9_. Furthermore, when we compared our serpentine adaptation candidates ($n = 61$) to candidate loci for parallel serpentine adaptation in _Arabidopsis lyrata_ ($n = 62$) from a previous study[47], we found two loci in common (significant overlap; $p < 0.007$), _KUP9_ and _TPC1_, further supporting important roles of these two ion transporters in repeated adaptation to serpentine soil. In addition, when overlapping the candidate genes detected at least in one of our

five population pairs with serpentine _A. lyrata_ study we revealed additional convergent loci involved in ion homoeostasis, calcium, nickel, and potassium transmembrane transport (Supplementary Table 5), suggesting existence of 'hotspot' regions in _Arabidopsis_ genome in response to serpentine stress. An additional candidate gene (_FPN2 = IREG2_) investigated in _Alyssum_ (Brassicaceae; Sobczyk et al.[48]) has been found to be shared between three population pairs. Finally, when comparing to the only genomically investigated serpentine system outside Brassicaceae

(*Mimulus*, Phrymaceae; Selby[49]), there was only limited overlap in two loci with one of our population pair. On the other hand, similar functions were enriched altogether suggesting parallel adaptation through similar pathways in very divergent (~140 myr) species.

SNP data only present part of the picture and, despite linkage, do not capture structural variation. Specific TE families can be activated by abiotic stresses and possibly contribute to adaptation to challenging environments[11,50–52]. Thus, we also investigated divergence at TEs in population pairs 1 to 4 (relatively lower coverage of the N5 population did not permit this analysis) based on 21,690 TE variants called using the TEPID approach that is specifically designed for population TE variation studies[53]. Assuming linkage between each TE variant and surrounding SNPs (in the proximity of ±100 bp), we applied a similar differentiation outlier window-based workflow as specified above and identified 92–115 TE-associated candidate genes per S–N contrasts (Supplementary Data 8). In comparison with the list of candidates based on SNPs (for the same four population pairs, $n = 1,853$), we observed the overlap of 46 genes. The GO enrichment of TE-associated candidates from all four population pairs resulted in significant enrichment ($p < 0.05$) of functions such as transmembrane transport, water channel activity and symporter activity (Supplementary Data 9). By overlapping the lists of TE-associated candidates across S–N pairs, we identified 13 parallel TE-associated differentiation candidates (Supplementary Fig. 9 and Supplementary Data 10; significant overlap, $p < 0.05$). These loci included the plasma membrane protein *PIP2*, the putative apoplastic peroxidase *PRX37*, and *RALF-LIKE 28*, which is involved in calcium signalling. This suggests a potential impact of TEs on serpentine adaptation and gives discrete candidates for future study.

**Sources of adaptive variation**. Next, we tested whether variants in each serpentine adaptation candidate have arisen by parallel de novo mutations or instead came from pre-existing variation shared across populations. To do so, we modelled allele frequency covariance around repeatedly selected sites for each locus and identified the most likely of the four possible evolutionary scenarios using a designated 'Distinguishing among Modes of Convergence' (DMC) approach[54]: (i) a null-model assuming no selection (neutral model), (ii) independent de novo mutations at the same locus, (iii) repeated sampling of shared ancestral variation, and (iv) sharing of adaptive variants via migration between adapted populations. For simplicity, we considered scenarios (iii) and (iv) jointly as 'shared origin' because both processes operate on alleles of a single mutational origin, in contrast to scenario (ii). To choose the best fitting scenario for each of the 61 candidate genes, we compared the maximum composite log-likelihoods (MCLs) between the four scenarios (see 'Methods'; Supplementary Table 6). This analysis indicated that parallel selection exceeded the neutral model for 62 out of the total 84 candidate cases of parallelism (i.e. cases when two population pairs shared one of the 61 serpentine adaptation candidates). To focus only on well-justified candidates of adaptation *within* the serpentine populations, we excluded an additional 33 cases where the scenario of parallel selection with the highest MCL estimate in serpentine populations was not considerably higher (>10%) than this estimate in non-serpentine populations, which resulted in 29 candidate cases of serpentine adaptation parallelism. Shared origins dominated these results, representing 97% of the cases (28/29; Fig. 2d and Supplementary Data 7). The alternate non-neutral scenario, parallel de novo origin, was supported only for a single locus, *TWO PORE CHANNEL 1* (*TPC1*) in one case (S3–N3 and S4–N4; Fig. 2e). Using a more permissive threshold for

identifying differentiation candidates (3% outliers, leading to a fivefold increase in parallel candidates) resulted in a similar DMC estimate of the proportion of the shared variation scenario (103/114 cases, i.e. 90%; Supplementary Fig. 10 and Supplementary Data 11), indicating that our inference of the dominant role of shared variation in genic parallelism is not dependent on a particular stringent outlier threshold. Finally, we applied a similar approach to parallel TE-associated differentiation candidates ($n = 13$) assuming selection on TE variants left a footprint in surrounding SNP–allele frequency covariance. We found a single non-neutral candidate, *ATPUX7*, for which parallel selection on standing variation was inferred (Supplementary Data 10). In summary, by a combination of genome-wide scanning with a designated modelling approach, we find that a non-random fraction of loci is likely reused by selection on serpentine, sourcing almost exclusively from a pool of alleles shared across the variable autotetraploid populations. Note that our conservative approach, focussed on identifying regions of repeated excessive differentiation and significant soil-related allele frequency differences, is not designed to cover the entire range of adaptive loci. Further research is thus needed to comprehensively cover the complete landscape of adaptation in autotetraploid *A. arenosa*.

**Rapid recruitment of convergent de novo mutations at the calcium channel *TPC1***. One advantage of the DMC approach is an objective model selection procedure. However, it does not give fine scale information about the distribution of sequence variation at particular alleles. Therefore, we further investigated candidate alleles of the *TPC1* gene, for which DMC results suggested the sweep of different de novo mutations in independent serpentine populations. Upon closer inspection of all short-read sequences complemented by Sanger sequencing of additional 40 individuals from the three serpentine populations, a remarkably specific selection signal emerged. We found two absolutely serpentine-specific, high-frequency, non-synonymous mutations only at residue 630, overlapping the region of the highest MCL estimate for the de novo scenario in DMC, and directly adjacent to the selectivity gate of the protein in structural homology models (Fig. 3 and Supplementary Table 7). Of the two, the polymorphism Val630Leu is nearly fixed in the S3 population (25 homozygous Leu630 individuals and four heterozygous Leu630/Val630 individuals out of 29 individuals) and is at a high frequency in the S5 population (one homozygous Leu630 individual and 17 heterozygous out of 20 individuals); the second convergent Val630Tyr mutation is at high frequency in the S4 population (four homozygous Tyr630 individuals and 18 heterozygous Tyr630/Val630 out of 25 individuals; Fig. 3a, b and Supplementary Fig. 11). Strikingly, Val630Tyr requires a three-nucleotide mutation covering the entire codon (GTA to TAT). Neither of these variants were found in any other *A. arenosa* population in a range-wide catalogue[8] encompassing 1724 *TPC1* alleles (including 368 alleles from the focal area of Eastern Alps; Fig. 3a) nor in the available short-read data of the other two *Arabidopsis* outcrossing species (224 *A. lyrata* and 178 *Arabidopsis halleri* alleles, respectively, Supplementary Fig. 12), indicating that both are private to serpentine populations. Altogether, the absolute lack of either *A. arenosa* serpentine-specific variant in non-serpentine sampling across the genus strongly supports the conclusions of the DMC modelling of their independent de novo mutation origin.

To investigate the potential functional impact of these high-frequency, convergent amino acid changes, we first generated an alignment for *TPC1* homologues across the plant and animal kingdoms. Residue 630 (634/633 in *Arabidopsis thaliana*/*A. lyrata*) is conserved as either a Val or Ile across kingdoms,

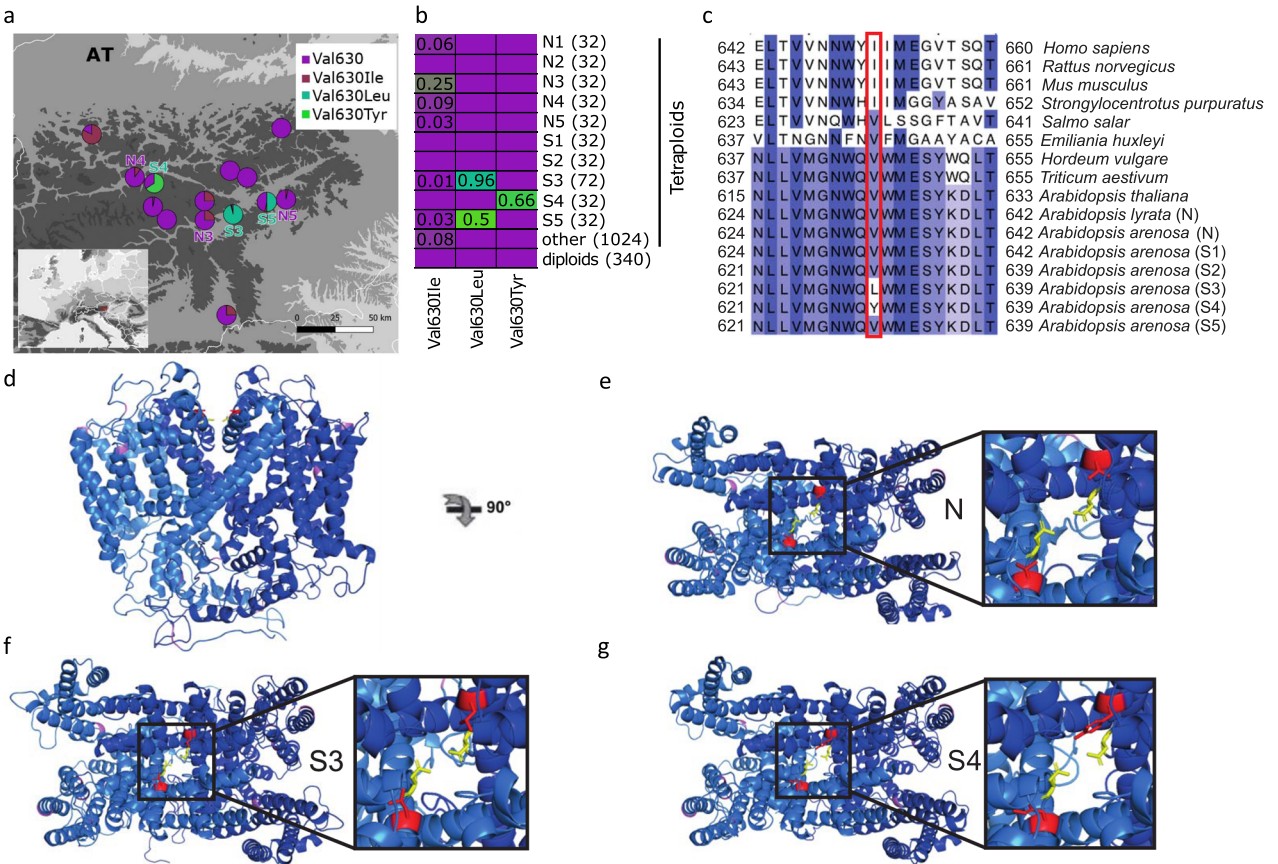

**Fig. 3 Serpentine-private, convergent de novo high-impact protein changes in the *TWO PORE CHANNEL 1* (*TPC1*) locus. a** All populations with serpentine-specific variants and all other resequenced *A. arenosa* populations in the focal area of Eastern Alps, showing frequencies of amino acid substitutions at residue 630 as pie charts (map drawn by V. Konečná). **b** Population frequencies of substitutions in the residue 630 among 1,724 alleles from range-wide *A. arenosa* resequenced samples. Colours denote frequencies from ancestral non-serpentine (violet) to serpentine-specific alleles (green) and number in brackets denotes total *N* of alleles screened. **c** Cross-kingdom conservation of the site shown by multiple sequence alignment of surrounding exon, including consensus sequences (AF > 0.5) from all serpentine *A. arenosa* populations (in S5 population, the frequency of Val630Leu is 0.5). Residues are coloured according to the percentage that matches the consensus sequence from 100% (dark blue) to 0% (white), the position of the serpentine-specific high-frequency non-synonymous polymorphism is highlighted in red. **d–g** Structural homology models of *A. arenosa TPC1* alleles. Dimeric subunits are coloured blue or marine. Non-synonymous variation that is not linked to serpentine soil is coloured deep purple. Residue 630 is coloured red and drawn as sticks. The adjacent residue, 627 (631 in *A. thaliana*), which has an experimentally demonstrated key role in selectivity control, is yellow and drawn as sticks. **d** Side view of the non-serpentine allele. **e** Top view of the non-serpentine allele with the detail of the pore opening depicted in the inset. **f** Top view of the Leu630 allele private for S3 and S5 populations. **g** Top view of the Tyr630 allele private for S4 population. Source data underlying Fig. 3a, b are provided as a Source data file.

except for the serpentine *Arabidopsis* populations (Fig. 3b, c and Supplementary Fig. 13). Val/Ile and Tyr are disparate amino acids in both size and chemical properties, so this substitution has high potential for a functional effect. Although the difference between Val/Ile and Leu is not as radical, we predicted that the physical difference between the side chains of Val/Ile versus Leu (the second terminal methyl group on the amino acid chain) may also have a functional effect by making new contacts in the tertiary structure. To test this, we performed structural homology modelling of all alleles found in *A. arenosa* (Fig. 3d–g), using two crystallographically determined structures as a template (PDB codes 5DQQ and 5E1J[55,56]). In the tertiary structure, residue 630 sits adjacent to the Asn residue (Asn627 in *A. arenosa*), which forms the pore's constriction point and has been shown to control ion selectivity in *A. thaliana*[55–58]. In *A. thaliana*, this Asn627 residue, when substituted by site-directed mutagenesis to the human homologue state, can cause Na$^+$ non-selective *A. thaliana TPC1* to adopt the Na$^+$ selectivity of human *TPC1*[56]. Depending on the rotameric conformation adopted, the Leu630 allele forms contacts with the selectivity-

determining Asn627 residue, while the non-serpentine Val630 does not (Fig. 3e).

Our modelling suggests that the Tyr630 residue is even more disruptive: the Tyr side chain can adopt one of the two broad conformations, either sticking into the channel, where it occludes the opening, or sticking away from the channel and directly into surrounding residues, which have been shown to be important for stabilising Asn627 in *A. thaliana*. Both of the conformations seen, shown in heterodimer form (Fig. 3g), are highly likely to disrupt the stability of Asn627, thereby modifying the selectivity of the channel. Finally, to determine likely rotameric conformations, we generated 100 models of the S4 homodimer (two Tyr630 alleles). In this case, the two residues were significantly less likely to both point into the channel (Supplementary Table 8), suggesting that this is difficult for the structure to accommodate. Because the S4 Tyr630 allele is predominantly in a heterozygous state with the Val630 allele in nature, we also generated 100 models of the S4 heterodimer (one Tyr630 and one Val630 allele). In the heterodimer, there was no significant difference between the occurrence of the two side-chain conformations (Supplementary

Table 8), suggesting that either disruptive conformation, sticking into the pore or affecting nearby functional residues, is possible. Taken together, these results suggest that even the presence of a single Tyr630 variant within a dimer will have a substantial impact on *TPC1* function and that the Tyr630 allele is likely to be dominant or partially dominant. This prediction is consistent with its predominant occurrence in a heterozygous state and a ~50% allele frequency in S4 (Supplementary Fig. 11). We also speculate that if homodimers of the Tyr allele are too disruptive to *TPC1* function this may result in a heterozygote advantage maintained by balancing selection. In conclusion, our results suggest that the serpentine-linked allelic variation at residue 630 impacts the selectivity of *TPC1*, with functional implications, and that independent convergent de novo mutations have been repeatedly selected upon during adaptation to serpentine soils. Dedicated, electrophysiological single vacuole conductance experiments are required to explore functional changes in detail at *TPC1*. However, the exceptionally suggestive convergent changes we discovered and modelled to structures at the pore selectivity gate force the speculation that they mediate change in the relative conductance of the divalent cations $Ca^{2+}$ and $Mg^{2+}$, the highly skewed ratios of which stand as hallmarks of serpentine soils[26].

## Discussion

Here we inferred the evolutionary sources of adaptive variation in autotetraploid populations by taking advantage of naturally replicated adaptation to toxic serpentine soil in wild *A. arenosa*. Using a designated statistical approach leveraging parallelism, we inferred that nearly all parallel serpentine adaptation candidates were sourced from a common pool of alleles that was shared across populations. However, for one exceptional candidate—a central calcium channel shown to mediate stress signalling[46]—we identified independent de novo mutations at the same otherwise highly conserved site with likely functional consequences. Our approach informs on natural sources of parallel adaptive variation in autotetraploid populations, which has not yet been investigated genome-wide in a natural polyploid system. Yet we refrain from direct comparison with diploid ancestors because diploid serpentine populations are not known in *A. arenosa*, leaving a space for further study investigating other species encompassing multiple ploidies facing the same environmental challenge.

The potential of polyploidy to enhance adaptation is a matter of ongoing debate that is mainly fuelled by theory-based controversies revolving around the efficiency of selection[6,19,59,60] and observations of the frequencies of WGD events in space and time[5,22]. In contrast, empirical population-level investigations that unravel evolutionary mechanisms operating in natural autopolyploids are very scarce. It has been shown that autopolyploids may rapidly react to new challenges by landscape genetic[23] and experimental studies[24,61]. Our transplant experiments coupled with demographic investigations support this and further demonstrate that such rapid adaptation may be repeated many times within a species, partially drawing on the same variants. Furthermore, a previous study in *A. arenosa* showed that the genome-wide proportion of non-synonymous polymorphisms fixed by directional selection was higher in tetraploids than in diploids, suggesting increased adaptive variation in natural autopolyploids[8]. However, it remains unclear whether such variation reflects increased input of novel mutations (as observed, e.g. in experimental yeast populations[17]) or sampling from increased standing variation (as predicted by theory[5,6]). Here we find nearly exclusive repeated sampling from a shared pool of variants. Such a dominant role of pre-existing variation is in line with the studies of parallel adaptation in diploid systems such as

*Littorina* snails[62,63], stickleback fishes[64,65], *Helliconius* butterflies[66], *Sinosuthora webbiana* vinous-throated parrotbill birds[39], *Ipomoea purpurea* morning glories[67], *Apis cerana* Asian honeybees[68], or *Coilia nasus* fishes[69]. Yet examples are lacking from autopolyploid systems, where a large pool of standing variation is expected by theory due to larger effective population size and polysomic masking of allelic variation[5,70]. Sharing alleles that have persisted in a specific genomic environment already for some time may be particularly beneficial under intense selection when rapid adaptive responses are needed[67,71–73]. In addition, standing genetic variation likely minimises negative pleiotropic effects of linked variants[67,74,75]. It should be noted, however, that our estimates may be biased upward for shared alleles by focussing only on cases of parallelism, which provided a testable framework for our inference of the sources of variants. Larger fractions of novel mutations may be represented among the non-parallel adaptive variation, which is, however, harder to identify.

In contrast to shared variants, empirical evidence for parallel de novo mutations within species is rare even in diploids[73,76–78] and we lack any example from polyploids. Theory suggests that such a scenario is unlikely for autopolyploids, as reduced efficacy of selection on a novel, initially low-frequency variants is predicted for most dominance states in autopolyploids[6,18,19]. On the other hand, beneficial alleles are introduced at increased rates in doubled genomes[6,17] and additional variation may accumulate due to polysomic masking[6]. Here we provide an example of parallel recruitment of two distinct de novo mutations with likely phenotypic effect in separate polyploid populations within one species, demonstrating that adaptive sourcing from novel polymorphisms is in fact feasible even in autopolyploids. Interestingly, high frequencies of homozygous individuals in one population demonstrates that such novel variants may approach fixation, in stark contrast to theory, which predicts incomplete sweeps of dominant mutations to be prevalent in autotetraploids[6,19]. On the other hand, the prevalence of heterozygotes in the other serpentine population together with results of structural modelling suggest (at least partial) dominance of the serpentine allele.[6]

Overall, our study demonstrates that rapid environmental adaptation may repeatedly occur in established autopolyploid populations. Footprints of selection at similar genomic positions mostly occur because of the repeated recruitment from a large pool of pre-existing variation, yet exceptionally also from recurrent de novo mutations. Thus, these results support the emerging view of autopolyploids as diverse evolutionary amalgamates, capable of flexible adaptation in response to environmental challenge.

## Methods

**Field sampling**. Serpentines occur in Central Europe as scattered edaphic 'islands' surrounded by open rocky habitats on other substrates in which autotetraploid *A. arenosa* frequently occur. In contrast, *A. arenosa* colonised only some serpentine sites in this area[79,80] indirectly suggesting that colonisation of serpentine sites by surrounding non-serpentine populations happened in parallel and was probably linked with local substrate adaptation. To test this hypothesis, we sampled all five serpentine (S) populations of *A. arenosa* known to date and complemented each by a proximal (<19 km distant) non-serpentine (N) population. All N populations grew in similar vegetation (rocky outcrops in open forests or grasslands) and soil type (siliceous to neutral rocks; Supplementary Table 1). Although we observed a considerable variation in the overall soil chemistry in our samples (Supplementary Fig. 14), the principal soil factors differentiating between S and N populations were always the same—higher Mg, Ni, and Co and lower Ca/Mg in S populations. Diploid serpentine *A. arenosa* is not known, even though serpentine barrens are frequent in some diploid-dominated areas, such as the Balkan peninsula. The sampled populations covered considerable elevational gradient (414–1750 m a.s.l.), but the differences in elevation within the pairs were small except for one pair where no nearby subalpine non-serpentine population exists (population pair S4–N4, difference 740 m).

We sampled eight individuals per every population for genomic analysis and confirmed their tetraploid level by flow cytometry. For each individual, we also sampled soil from very close proximity to the roots (~10–20 cm below ground),

except for N5 population for which genotyped data were already taken from the previous study[8]. There we collected an additional eight soil samples and use their average in the following environmental association analysis. For the transplant experiment, we also collected seeds (~20–30 maternal plants/population) from three population pairs (S1–N1, S2–N2, S3–N3) and bulks of soil ~80 l (sieved afterwards) from the natural sites occupied by these six populations.

**DNA extraction, library preparation, sequencing, raw data processing, and filtration**. We stored all samples for this study in RNAlater (R0901-500ML, SIGMA-ALDRICH CO LTD) to avoid genomic DNA degradation and we further prepared the leaf material as described in refs. [81–83]. We extracted DNA as described in Supplementary Method 1. Genomic libraries for sequencing were prepared using the Illumina TRUSeq PCR-free library. Libraries were sequenced as 150 bp paired-end reads on a HiSeq 4000 (3 lanes in total) by Norwegian Sequencing Centre, University of Oslo.

We used trimmomatic-0.36[84] to remove adaptor sequences and low-quality base pairs (<15 PHRED quality score). Trimmed reads >100 bp were mapped to reference genome of North American *A. lyrata*[85] by bwa-0.7.15[86] (https://rcc.uchicago.edu/docs/software/modules/bwa/midway2/0.7.15.html) with default setting. Duplicated reads were identified by picard-2.8 (https://github.com/broadinstitute/picard) and discarded together with reads that showed low mapping quality (<25). Afterwards we used GATK v.3.7 to call and filter reliable variants and invariant sites according to best practices[87] (complete variant calling pipeline available at https://github.com/vlkofly/Fastq-to-vcf). Namely, we used the HaplotypeCaller module to call variants per individual using the ploidy = 4 option, which enables calling full tetraploid genotypes. Then we aggregated variants across all individuals by module GenotypeGVCFs. We selected only biallelic SNPs and removed those that matched the following criteria: Quality by Depth (QD) < 2.0, FisherStrand (FS) > 60.0, RMSMappingQuality (MQ) < 40.0, MappingQualityRankSumTest (MQRS) < −12.5, ReadPosRankSum < −8.0, StrandOddsRatio (SOR) > 3.0. We called invariant sites also with the GATK pipeline similarly to variants, and we removed sites where QUAL was <15. Both variants and invariants were masked for sites with average read depth (RD) >2 times standard deviation as these sites were most likely located in duplicated regions and we also masked regions with excessive heterozygosity, representing likely paralogous mis-assembled regions, following Monnahan et al.[8]. One individual per each S2 and N5 populations was excluded due to exceptionally bad data quality (low percentage of mapped reads and low RD, <10 on average), leaving us with a final data set of 78 individuals that were used in genomic analyses. This pre-filtered data set contained 110,358,565 sites (of which 11,744,200 were SNPs) with average depth of coverage 21× (Supplementary Data 1). This way of genotyping leads to allele frequency estimates that are well comparable with previous estimates[88] including a broad range-wide *A. arenosa* sampling[8] (the site frequency spectra (SFS) are presented in Supplementary Fig. 15). Our site frequency estimates, which were constructed by program est-sfs[89], are likely not biased by the number of individuals as was demonstrated by consistent site frequency estimates when subsampling one more deeply sampled population to the most common number of 32 chromosomes (S3; when including additional nine individuals from Arnold et al.[29]; Supplementary Fig. 16).

**Reconstruction of population genetic structure**. We inferred the population genetic structure, diversity, and relationships among individuals from putatively neutral 4dg SNPs filtered for DP >8 per individual and maximum fraction of filtered genotypes (MFFG) of 0.2, i.e. allowing max. 20% missing calls per site (1,042,793 SNPs with a total of 0.49% missing data; see Supplementary Tables 1–3 and Supplementary Data 1 for description of data sets and filtration criteria). We used several complementary approaches. First, we ran principal component analysis (PCA) on individual genotypes using glPCA function in adegenet v.2.1.1 replacing the missing values by average allele frequency for that locus. Second, we applied model-based clustering with accelerated variational inference in fastStructure v.1.0[90]. To remove the effect of linkage, we randomly selected one SNP per a 1 kb window, keeping 10 kb distance between the windows and, additionally, filtered for minimum minor allele frequency (MAF) = 0.05 resulting in a data set of 9,923 SNPs. As fastStructure does not handle the polyploid genotypes directly, we randomly subsampled two alleles per each tetraploid site using a custom script. This approach has been demonstrated to provide unbiased clustering in autotetraploid samples in general[91] and *Arabidopsis* in particular[8]. We ran fastStructure with 10 replicates under K = 5 (corresponding to the number of population pairs) with default settings. Third, we inferred relationships among populations using allele frequency covariance graphs implemented in TreeMix v.1.13[92]. We used custom python3 scripts (available at https://github.com/mbohutinska/TreeMix_input) to create the input files. We ran TreeMix analysis rooted with an outgroup population (tetraploid *A. arenosa* population 'Hranovnica' from the area of origin of the autotetraploid cytotype in Western Carpathians[93]). We repeated the analysis over the range of 0–6 migration edges to investigate the change in the explanatory power of the model when assuming migration event(s) and found that adding migration to the model did not lead to large improvement (Supplementary Fig. 17). We bootstrapped the scenario without migration (the topology did not change with adding the migrations) choosing bootstrap block size 1 kb (the same window size also for the divergence scan, see below) and 100 replicates and

summarised the results using *SumTrees.py* function in DendroPy[78]. Finally, we calculated nucleotide diversity ($\pi$) and Tajima's D for each population and pairwise differentiation ($F_{ST}$) (Supplementary Table 9) for each population pair using custom python3 scripts (available at https://github.com/mbohutinska/ScanTools_ProtEvol; see Supplementary Table 3 for the number of sites per each population). For nucleotide diversity calculation, we down-sampled each population to six individuals on a per-site basis to keep equal sample per each population while also keeping the maximum number of sites with zero missingness.

**Demographic inference**. We performed demographic analyses in fastsimcoal v.2.6[94] to specifically test for parallel origin of serpentine populations and to estimate divergence time between serpentine and proximal non-serpentine populations. We constructed unfolded multidimensional SFS from the variant and invariant 4dg sites (filtered in the same ways as above, Supplementary Table 2) using custom python scripts published in our earlier study (FSC2input.py at https://github.com/pmonnahan/ScanTools/)[8]. We repolarized a subset of sites using genotyped individuals across closely related diploid *Arabidopsis* species to avoid erroneous inference of ancestral state based on a single reference *A. lyrata* individual following Monnahan[8].

First, we tested for parallel origin of serpentine populations using population quartets (two pairs of geographically proximal serpentine and non-serpentine populations) and iterated such pairs across all combinations of regions (10 pairwise combinations among the five regions in total). For each quartet, we created four-dimensional SFS and compared following the four evolutionary scenarios (Supplementary Fig. 2): (i) parallel origin of serpentine ecotype—sister position of serpentine and non-serpentine populations within the same region, (ii) parallel origin with migration—the same topology with additional gene flow between serpentine and the proximal non-serpentine population, (iii) single origin of serpentine ecotype—sister position of serpentine populations and of non-serpentine populations, respectively, and (iv) single origin with migration—the same topology with additional gene flow between serpentine and the proximal non-serpentine populations. For each scenario and population quartet, 50 fastsimcoal runs were performed. For each run, we allowed for 40 ECM optimisation cycles to estimate the parameters and 100,000 simulations in each step to estimate the expected SFS. We used wide range of initial parameters (effective population size, divergence times, migration rates; see the example *.est and *.tpl files provided for each model tested in the Supplementary Data 12) and assumed mutation rate of $4.3 \times 10^{-8}$ inferred for *A. arenosa* previously[32]. Further, we extracted the best likelihood partition for each fastsimcoal run, calculated Akaike information criterion (AIC), and summarised the AIC values across the 50 fastsimcoal runs. The scenario with lowest median AIC values within each particular population quartet was preferred (Supplementary Fig. 2 and Supplementary Data 2).

Second, we estimated divergence time between S and N populations from each population pair (i.e. S1–N1, S2–N2, S3–N3, S4–N4, and S5–N5) based on two-dimensional SFS using the same fastsimcoal settings as above. We simulated according to models of two-population split, not assuming migration because the model with migration did not significantly increase the model fit across the quartets of populations (Supplementary Fig. 2 and Supplementary Data 2). To calculate 95% confidence intervals for parameter estimates (Table 1), we sampled with replacement the original SNP matrices to create 100 bootstrap replicates of the two-dimensional SFS per each of the five population pairs.

**Window-based scans for directional selection**. We leveraged the fivefold-replicated natural set-up to identify candidate genes that show repeated footprints of selection across multiple events of serpentine colonisation. First, we identified genes of excessive divergence for each pair of proximal serpentine (S)–non-serpentine (N) populations (five pairs in total). Reflecting hierarchical structure in the data, we avoided merging multiple populations into larger units for the estimation of $F_{ST}$ and strictly worked in a pairwise design. We admit that population S3 occupies a somewhat separate position and its ancestral non-serpentine population might have thus remained unsampled (or got extinct)—we therefore used the geographically closest non-serpentine population N3 as the most representative ancestral non-serpentine population available in our sampling. We calculated pairwise $F_{ST}$[95] for non-overlapping 1 kbp windows along the genome with the minimum of 10 SNPs per window to exclude potential biases in $F_{ST}$ estimation caused by low-informative windows[96]. We used the custom script (https://github.com/mbohutinska/ScanTools_ProtEvol) based on ScanTools pipeline that was successfully applied in our previous analyses of autotetraploid *A. arenosa*[8]. The window size of 1 kbp was selected to properly account for the average genome-wide linkage disequilibrium (LD) decay of genotypic correlations (150–800 bp) previously estimated in autotetraploid *A. arenosa*[40]. We used windows of fixed length and thus with homogeneous position on genome across all population pairs (in contrast to windows defined by number of SNPs and thus varying in exact position and length) to facilitate comparisons of selection candidates across distinct population pairs. Our $F_{ST}$ estimates are unlikely to be strongly affected by varying numbers of SNPs per window, as the correlation between $F_{ST}$ and number of SNPs per window was very weak (Spearman's rank correlation coefficient varied from 0.02 to 0.04 across population pairs; Supplementary Fig. 18). We identified the upper 99% quantile of all 1 kbp windows in the empirical distribution of $F_{ST}$ metric per each population pair. Then we identified

initial lists of genes with excessive differentiation for each S–N population pair as genes overlapping with the 1% outlier windows using *A. lyrata* gene annotation[97], where the gene includes 5' untranslated regions (UTRs), start codons, exons, introns, stop codons, and 3' UTRs. Then we refined this inclusive list and identified parallel differentiation candidates as genes identified as candidates in at least two population pairs. We tested whether such overlap is higher than a random number of overlapping items given the sample size using Fisher's exact test in Super-ExactTest R package[98]. Our $F_{ST}$-based detection of outlier windows was not largely biased towards regions with low recombination rate (based on the available *A. lyrata* recombination map[99]; Supplementary Fig. 19).

**Environmental association analysis**. To further refine the candidate list to genes associated with the discriminative serpentine soil parameters, we performed environmental association analysis using LFMMs—LFMM 2 (https://bcm-uga.github.io/lfmm/)[41]. We tested the association of allele frequencies at each SNP for each individual with associated soil concentration of the key elements differentiating serpentine and non-serpentine soils: Ca/Mg ratio and bioavailable soil concentrations of Co, Mg, and Ni. Only those elements were significant in one-way ANOVAs (Bonferroni corrected) testing differences in elemental soil concentration between S and N population, taking population pair as a random variable. We retained 1,783,055 SNPs without missing data and MAF > 0.05 as an input for the LFMM analysis. LFMM accounts for a discrete number of ancestral population groups as latent factors. We used five latent factors reflecting the number of population pairs. Due to hierarchical structure in the data (PCA based on ~1 M 4dg SNPs indicated five main groups, yet the first three axes alone also explained considerable variation; Supplementary Fig. 20), we also performed the additional analysis assuming three latent factors. As such analysis had only a minor effect on the total number of serpentine adaptation candidates (reducing their number by only two), we further used a candidate list based on five latent factors that corresponds to the total number of population pairs, thus it is also better comparable with the parallel differentiation candidates. To identify SNPs significantly associated with soil variables, we transformed $p$ values to false discovery rate (<0.05) based on $q$ values using the qvalue R package v.2.20[100]. Finally, we annotated the candidate SNPs to genes, termed 'LFMM candidates' (at least one significantly associated SNP per candidate gene).

We made a final shortlist of serpentine adaptation candidates by overlapping the LFMM candidates, reflecting significant association with important soil elements, with the previously identified parallel differentiation candidates, mirroring regions of excessive differentiation repeatedly found across parallel population pairs. For visualisation purposes (Fig. 2e), we annotated SNPs in the serpentine adaptation candidates using SnpEff v. 4.3[101] following *A. lyrata* version 2 genome annotation[97].

**TE variant calling and analysis**. TE variants (insertions or deletions) among sequenced individuals were identified and genotyped in population pairs 1–4 using TEPID v.0.8[53] following the approach described in Rogivue et al.[50] (relatively lower coverage of the N5 population did not permit this analysis in the last pair). We annotated TEs based on available *A. lyrata* TE reference[102]. TEPID is based on split and discordant read mapping information and employs read mapping quality, sequencing breakpoints, and local variation in sequencing coverage to call the absence of reference TEs as well as the presence of non-reference TE copies. This method is specifically suited for studies at the population level as it takes intra-population polymorphism into account to refine TE calls in focal samples by supporting reliable call of non-reference alleles under lower thresholds when found in other individuals of the population. We filtered the data set by excluding variants with MFFG > 0.2 and DP < 8, which resulted in 21,690 TE variants (13,542 deletions and 8,148 insertions as compared to the reference).

Assuming linkage between TE variants and nearby SNPs[53], we calculated pairwise $F_{ST}$[95] using SNP frequencies (in the same way as specified above) for non-overlapping 1 kbp windows containing TE variant(s) for each population pair. The candidate windows for directional selection were identified as the upper 99% quantile of all windows containing a TE variant in the empirical distribution of $F_{ST}$ metric per each population pair. Further, we identified candidate genes (TE-associated candidates) as those present up to +/−2 kbp upstream and downstream from the candidate TE variant (assuming functional impact of TE variant until such distance, following Hollister et al.[103]). Finally, we identified parallel TE-associated differentiation candidates as those loci that appeared as candidates in at least two population pairs.

**GO enrichment analysis**. We inferred potential functional consequences of the candidate gene lists using GO enrichment tests within biological processes, molecular functions, and cellular components domains. We applied Fisher's exact test ($p < 0.05$) with the 'elim' algorithm implemented in topGO v.2.42 R package[104]. We worked with *A. thaliana* orthologues of *A. lyrata* genes obtained using biomaRt[105] and *A. thaliana* was also used as the background gene universe in all gene set enrichment analyses. The used 'elim' algorithm traverses the GO hierarchy from the bottom to the top, discarding genes that have already been mapped to significant child terms while accounting for the total number of genes annotated in the GO term[104,106].

**Modelling the sources of adaptive variation**. For each serpentine adaptation candidate ($n = 61$ and $n = 13$ identified using SNPs and TE-associated variants, respectively), we modelled whether it exhibits patterns of parallel selection that is beyond neutrality and if so whether the parallel selection operated on de novo mutations or rather called on pre-existing variation shared across populations. We used model-based likelihood approach that is specifically designed to identify loci involved in parallel evolution and to distinguish among their evolutionary sources (DMC[54]). Convergent is analogous to parallel in this case as the entire approach is designed for closely related populations.

We considered the following four evolutionary scenarios assuming distinct variation sources: (i) no selection (neutral model), (ii) independent de novo mutations at the same locus, (iii) repeated sampling of ancestral variation that was standing in the non-adapted populations prior the onset of selection, and (iv) transfer of adaptive variants via gene flow (migration) from another adapted population. We interpreted the last two scenarios together as a variation that is shared across populations as both operate on the same allele(s) that do not reflect independent mutations.

We estimated composite log-likelihoods for each gene and under each scenario using a broad range of realistic parameters taking into account demographic history of our populations inferred previously and following recommendations in Lee and Coop[54] (positions of selected sites, selection coefficients, migration rates, times for which allele was standing in the populations prior to onset of selection, and initial allele frequencies prior to selection; see Supplementary Table 6 for the summary of all parameters and their ranges). We chose to place selected sites at eight locations at equal distance (default value recommended by authors https://github.com/kristinmlee/dmc) from each other along the particular gene. Such a density (one site per ~500 bp on average, as the mean length of serpentine adaptation candidate is ~4,000 bp) is in fact well within the range of the LD decay of 150–800 bp, which was estimated in *A. arenosa*[40]. For the calculation of co-ancestry decay, we also considered a 25 kbp upstream and downstream region from each gene. To choose the best fitting scenario for each candidate, we first estimated the MCL over the parameters for each of the three parallel selection scenarios and a neutral scenario. We selected among the parallel selection models by choosing the model with the highest MCL, following the approach of ref. [49]. Further, we considered the case significantly non-neutral only if the MCL difference between the selected parallel model and the corresponding neutral model was higher than the maximum of the distribution of the differences from the simulated neutral data in *A. arenosa* inferred in Bohutínská et al.[40] (i.e. MCL difference >21, a conservative estimate). Further, to focus only on divergence caused by selection in serpentine populations (i.e. eliminating divergence signals caused by selection in N populations), only cases of selection in the serpentine populations in which the scenario of parallel selection had a considerably higher MCL estimate (>10%) than selection in non-serpentine populations were taken into account.

To ensure that our inference on the relative importance of shared versus de novo variation is not biased by arbitrary outlier threshold selection, we re-analysed the SNP data set using a 3% outlier $F_{ST}$ threshold for identifying differentiation candidates. We overlapped the resulting 1,179 parallel differentiation candidates with the LFMM candidates and subjected the resulting 246 serpentine adaptation candidates to DMC modelling in the same way as described above (Supplementary Data 11). For each of the 246 genes and parallel quartets of populations (in total 420 cases of parallelism) we again compared the four scenarios as described above (Supplementary Fig. 10 and Supplementary Data 11).

**Reciprocal transplant experiment**. To test for local substrate adaptation in three serpentine populations, we compared plant fitness in the native versus foreign soil in a reciprocal transplant experiment. We reciprocally transplanted plants of serpentine and non-serpentine origin from three population pairs (S1–N1, S2–N2, S3–N3) that served as representatives of independent colonisation in each broader geographic region (Bohemian Massif, lower Austria, and Eastern Alps, respectively). As the most stressful factor for *A. arenosa* populations growing on serpentine sites is the substrate (Arnold et al.[29] and Supplementary Fig. 3a, b), we isolated the soil effect by cultivating plants in similar climatic conditions in the greenhouse.

For each pair, we cultivated the plants in serpentine and non-serpentine soil originating from their original sites (i.e. S1 plant cultivated in S1 and N1 soil and vice versa) and tested for the interaction between the soil treatment and soil of origin in selected fitness indicators (germination and rosette diameter sizes). We germinated seeds from 12 maternal plants (each representing a seed family of a mixture of full- and half-sibs) from each population in Ppetri dishes filled by either type of soil (15 seeds/family/treatment). Seeds germinated in the growth chamber (Conviron) under conditions approximating spring season at the original sites: 12 h dark at 10 °C and 12 h light at 20 °C. We recorded the germination date as the appearance of cotyledon leaves for the period of 20 days after which there were no new seedlings emerging. We tested for the effect of substrate of origin (serpentine versus non-serpentine), soil treatment (serpentine versus non-serpentine), and their interaction on germination proportion using GLM with binomial errors. To account for lineage-specific differences between population pairs, which are uninformative for the overall assessment of the fitness response towards serpentine, we treated population pairs as a random variable.

Due to zero germination of N1 seeds in S1 soil, we measured differential growth response on plants that were germinated in the non-serpentine soils and were subjected to the differential soil treatment later, in a seedling stage. We chose 44–50 seedlings equally representing progeny of 11 maternal plants per each population (in total 284 seedlings), transferred each plant to a separate pot filled either with ~1 L of the original or the alternative paired soil (i.e. S1 soil for N1 population and vice versa). We randomly swapped the position of each pot twice a week and watered them with tap water when needed. We measured the rosette diameter and counted the number of leaves (which correlated with the rosette diameter, $R = 0.85$, $p < 0.001$) twice a week for five weeks until rosette growth reached a plateau (Supplementary Fig. 6). By that time, we observed zero mortality and only negligible flowering (1%, four of the 284 plants). We tested whether soil treatment (serpentine versus non-serpentine) with the interaction of soil of origin (serpentine versus non-serpentine) had a significant effect on rosette diameter sizes (the maximum rosette diameter sizes from the last tenth measurement as a dependent variable), using two-way ANOVA taking population pair (1–3) as a random factor.

**Elemental analysis of soil and leaf samples**. We quantified the soil elemental composition by inductively coupled plasma mass spectrometry (ICP-MS; Perkin-nElmer NexION 2000, University of Nottingham). We monitored 21 elements (Na, Mg, P, S, K, Ca, Ti, Cr, Mn, Fe, Co, Ni, Cu, Zn, As, Se, Rb, Sr, Mo, Cd, and Pb) in the soil extract samples. Individual soil samples of genotyped individuals (80 samples in total) were dried at 60 °C. Soil samples were sieved afterwards. Samples were prepared according to a protocol summarised in Supplementary Method 2. We quantified the elemental soil and leaf composition of the elements in samples from reciprocal transplant experiment by ICP OES spectrometer INTE-GRA 6000 (GBC, Dandenong Australia). We monitored three elements that were identified as key elements differentiating S and N soils of the natural populations (Ca, Mg, and Ni) and decomposed samples prior the analysis. For details, see Supplementary Method 3.

**Screening natural variation in the *TPC1* locus**. To screen a broader set of *TPC1* genotypes in the relevant serpentine populations S3–S5, we Sanger-sequenced additional individuals sampled at the original sites of the focal S3, S4, and S5 populations (11, 17, and 12 individuals, respectively) exhibiting non-synonymous variation in the *TPC1* locus. For the amplification of the exon around the candidate site, we used specifically designed primers (Supplementary Table 10). The mix for PCR contained 0.3 µL of forward and reverse primer each, 14.2 µL of ddH2O, 0.2 µL of MyTaq DNA polymerase, and 4 µL of reaction buffer MyTaq, and we added 1 µL (10 ng) of DNA. The PCR amplification was conducted in a thermo-cycler (Eppendorf Mastercycler Pro) under the following conditions: 1 min of denaturation at 95 °C, followed by 35 cycles: 20 s at 95 °C, 25 s at 60 °C, 45 s at 72 °C, and a final extension for 5 min at 72 °C. Amplification products of high purity were sequenced at 3130xl Genetic Analyser (DNA laboratory of Faculty of Science, Charles University, Prague).

Then we checked whether the candidate alleles, inferred as serpentine specific in our data set, are also absent in a published broad non-serpentine sampling among outcrossing *Arabidopsis* species[8,33,107–112]. We downloaded all the available short-read genomic sequences published with the referred studies, called variants using the same approach as described above, and checked the genotypes at the candidate site (residue 630 in *A. arenosa* and 633 in *A. lyrata* and *A. halleri*). In total, we screened 1724 alleles of *A. arenosa*, 178 alleles of *A. halleri*, and 224 alleles of *A. lyrata*.

To visually compare variation at the entire *TPC1* locus, we generated consensus sequences for the group of all five non-serpentine *A. arenosa* populations and for each separate serpentine population using the Variant Call Format (VCF) with all variants in the region of scaffold_6:23,042,733–23,048,601 using bcftools (Supplementary Fig. 21). Sites were included in the consensus sequence if they had AF > 50%. As the original VCF contained only biallelic sites, an additional multiallelic VCF was also created using GATK and any variants with AF > 50% were manually added to the biallelic consensus sequence. The *A. lyrata* and *A. thaliana* non-serpentine sequences were assumed to match the corresponding reference for each species.

Finally, we screened the variation in the *TPC1* locus at deep phylogenetic scales. We generated multiple sequence alignments using Clustal-Omega[113] from the available mRNA sequences from GenBank (*Emiliania huxleyi*, *Hordeum vulgare*, *A. thaliana*, *Triticum aestivum*, *Salmo salar*, *Strongylocentrotus purpuratus*, *Homo sapiens*, *Rattus norvegicus*, and *Mus musculus*) that were complemented by the consensus *Arabidopsis* sequences described above. Alignments were manually refined and visualised in JalView[114].

**Structural homology models**. Structural homology models of dimeric TPC1 were generated with Modeller v. 9.24[115] using two *A. thaliana* crystal structures (5E1J and 5DQQ[55,57]) as templates. The final model was determined by its discrete optimised protein energy score.

**Reporting summary**. Further information on research design is available in the Nature Research Reporting Summary linked to this article.

## Data availability

Data supporting the findings of this work are available within the paper and its Supplementary Information files. A Reporting Summary for this Article is available as a Supplementary Information file. Sequence data generated in this study have been deposited in the GenBank SRA database as a BioProject PRJNA667586 (populations S1–S5, N1–N4). Additional sequence data used in this study are deposited in the GenBank SRA database within a BioProject PRJNA325082 (population N5). Source data are provided with this paper.

## Code availability

Newly developed scripts are available at GitHub [https://github.com/vlkofly/Fastq-to-vcf] and [https://github.com/mbohutinska/ScanTools_ProtEvol].

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

## Acknowledgements

This work was supported by the Czech Science Foundation (project 20-22783S to F.K.), Charles University (project Primus/SCI/35 to F.K and Charles University Grant Agency grant No. 410120 to V.K.), and the European Research Council (ERC) under the European Union's Horizon 2020 research and innovation programme [grant number ERC-StG 679056 HOTSPOT to L.Y. and ERC-StG 850852 DOUBLE ADAPT to F.K.]. Additional support was provided by the long-term research development project No. RVO 67985939 of the Czech Academy of Sciences. Computational resources were provided by the CESNET LM2015042 and the CERIT Scientific Cloud LM2015085. The authors thank Lenka Flašková, Gabriela Šrámková, Anna Krejčová, and Mellieha Allen for help with laboratory work and result interpretation.

## Author contributions

F.K., L.Y., and V.K. conceived the study. V.K., A.B.-G., L.Y., F.K., and M.B. performed field collections. A.B.-G., P.F., and V.K. did laboratory work. V.K., S.B., J.V., R.R.C., and M.B. performed analyses. V.K. and D.P. performed experiments. V.K., L.Y., and F.K. wrote the manuscript with input from all authors. All authors approved the final manuscript.

## Competing interests

The authors declare no competing interests.
