## [Peer Review File · Nature Communications]

REVIEWER COMMENTS

Reviewer #1 (Remarks to the Author):

This manuscript studied the genetic basis of parallel adaptation or evolution using five serpentine/non-serpentine populations pairs of autotetraploid *Arabidopsis arenosa*. Based on the selection scan and modeling analysis of the population genomics data, authors discovered substantial parallelism in candidate genes involved in ion homeostasis. In summary, this study suggested that polyploid populations can rapidly adapt to stress environment using both standing and novel genetic variation. Overall, this is a generally interesting study.

1) Figure 2a, apparently, no any differentiation gene shared across all the five pairs, is there any possible reason for this.

2) For Figure 2b, the GO enrichment was showed as molecular function, it is better to show biological processes as well.

3) For TE-associated differentiation candidate genes, it is not clear to what extent these differentiated candidate genes overlapped with the differentiated genes based on SNP. It is not mentioned in the manuscript.

Reviewer #2 (Remarks to the Author):

This paper claims to show, using genome sequence data from 78 plants from 10 populations, that independent adaptation to serpentine soils by different populations of the autotetraploid plant *Arabidopsis arenosa* involved "substantial parallelism." By this the authors mean that overlapping sets of candidate genes experienced positive selection in two or more of the five serpentine populations. The paper further claims that most instances of parallel adaptive evolution in candidate genes involved the selective recruitment same genetic variants across populations, rather than parallel adaptation via separate new mutations. However the authors do convincingly document a single case of separate new adaptive serpentine mutations in a TPC1 gene that appear, via protein modeling, to disrupt an ion channel.

The whole paper is framed around the idea that autopolyploid species may harbor more "standing" genetic variation than diploids because there are more gene copies per individual (4 vs 2), and so parallel adaptation via shared variants might be an especially prevalent mode of evolution in species like *A. arenosa*.

While I was initially quite interested and impressed by these claims and the framing of the paper, the closer I examined the underlying evidence the less convinced I was that the data and its analysis convincingly upholds the conclusions. Below I outline my main concerns.

1. I was not convinced that the framing of the paper, in the introductory and concluding paragraphs, around issues of diploid vs polyploid patterns of parallel adaptation was useful or particularly relevant to this study. First, is it really established that autotetraploids actually tend to have greater nucleotide diversity than their diploid progenitors? I could not find any such pattern among the papers cited by the authors. In fact the authors' earlier paper comparing diversity between diploid and polyploid *A. arenosa* found no different in pairwise diversity and only very slight increase of non-synonymous segregating site diversity but no difference insect synonymous diversity. As the authors pointed out, the young age of this autopolyploid might contribute to this lack of a pattern. And in fact the diversity levels for this autopolyploid given in Table 1 are right in line with those of many diploid outcrossing plant species. (question - are those values for π only for 4-fold degenerate sites? if so that should be stated clearly in the table)

2. The authors collected ~8 plants from 5 pairs of adjacent serpentine and non-serpentine

populations, and then for most of the analyses of F_{st} , LFMM, and DMC these 5 pairs were treated as if they were essentially independent of each other. However the PCA in Fig 1 clearly indicates that based on genome wide patterns of variation there really are only 3 main groupings of populations that are clearly divergent from each other (and the two populations in pair 3 clearly do not cluster together). The authors need to justify, on population genetic terms, their use of 5 pairs rather than 3 groups, and explain how this choice may influence their downstream conclusions about selection.

3. I am quite concerned about the low number of sequenced plants per population and how this might reduce statistical power and increase noise and false positives in the population genetic analyses of selection. Each plant has 4 copies of each site in the genome, and so there are a total of 5 possible genotypes at each bi-allelic SNP. It sounds like for each SNP these genotypes were called for each plant, and then population allele frequencies were estimated by essentially counting up the numbers of each allele in each of the 8 genotypes in each population - ideally each population allele frequency would be based on 32 total copies of each SNP. With a mean of 21x per site per individual, but with plenty of variation around this mean, how confident are we in the individual genotype calls and the resulting estimates of population allele frequencies given this variation of coverage, the 4 copy per site per individual issue, and the low number of individuals/allele counts per population? As far as I can tell all analyses take the allele frequencies estimates at face value, and they do not take into account all of the different sources of uncertainty.

4. I found the section outlining the F_{st} scans for each of the 5 pairs of populations to be a bit perplexing. I have several issues/questions. First, it would be helpful to see the Manhattan plots for individual SNPs in the supplemental figures as well as that for the windowed data - this would help the reader to see how noisy or clear the data look. Second, why window based on 1kb windows rather than on windows of fixed # of SNPs? And is that 1kb of genome length, or 1 kb of sites that are not masked? Because SNP density is expected to vary tremendously, windows based on physical length may "overly flatten" peaks if there are a lot of SNPs or they may not be included if they are in a conserved region with < 10 SNPs. There's no obvious best way to approach this, but it might be worth seeing what windows of 5, 10, 20, etc SNPs look like compared to the 1 kb windows. Finally, it seems worth looking at additional comparisons of F_{st} for serpentine vs non serpentine for other groupings of populations other than just the five pairs. For example, it seems compelling based on the PCA to group populations into the 3 clusters and examine F_{st} by habitat within each of those groups. And it certainly is worth investigating what happens to F_{st} by habitat if all populations are combined (calculating allele frequencies for each habitat using all plants).

5. Continuing with the F_{st} based scans, I found the evidence for parallel adaptation at candidate genes to be weak and outlined in the paper in a somewhat misleading way. Comparing the peaks in the plots of windowed F_{st} across the genome for each pair, it is immediately obvious that very few clear peaks are shared across populations. This may very well be due to low power or windowing issues to some extent but lets assume that its not for the time being. The authors find about 500 candidate genes have F_{st} window values in the top 1% in each population pair. Amazingly if you ask whether the 500 in any particular pair are also found in another pair, only about 20-40 are shared, meaning that > 900 candidate genes found in just one of the 2 pairs. The situation is much less compelling if you pick 3 or 4 or 5 of the 5 pairs and ask how many of the candidate genes show high F_{st} in those populations - just a couple of genes are shared in the 3 or 4 pairs and NONE of the 2,245 candidate genes show high F_{st} in all 5 populations. Taken at face value this is a remarkably low level of parallelism - the overwhelming pattern is of unique adaptations in each serpentine populations. This point is not clearly conveyed in the paper. Now, do we really believe in that exceedingly low level of parallelism? I strongly suspect that the small sample size/high noise/low power issue raised above causes both false positives (perhaps many of the ~500 candidate genes per pair are not actually under divergent selection but by chance F_{st} is estimated to be high) and false negatives (noise leads to underestimates of F_{st} in regions that are actually experiencing divergent selection).

6. I do not fully understand the LFMM 2 analyses and whether this approach is a powerful method for

identifying parallel adaptation or not. The paper does not clearly state that the unit of observation for this analysis is the individual and not the population, but because each of the 78 plants has genotype data at each SNP and results of chemical analyses of the soil around its roots during collection in the field, the authors attempt to use LFMM 2 to find associations between alleles at SNPs and soil data, while attempting to control or model into the analysis associations that might be caused by population structure alone. The problem is that the population structure in this collection of 10 populations is complex and hierarchical, and very clearly not simply 5 separate/independent pairs of populations (based on PCA and additional measures of structure in Supp. figs). And yet the latent factors used in this analysis to somehow model this structure is simply the 5 population pairs. To what extent would the same number and identify of genes be identified if other latent factors were used that perhaps more accurately account for structure - for example what about using instead those 3 broad clusters in PCA space? The authors need to clearly explain how these issues of population structure may or may not have influenced the results. In addition it was not clear to me whether this method should in principle only identify broadly parallel adaptations, since I imagine that it would not flag a gene if it was only divergent in one of the serpentine populations? And yet there is again remarkably little overlap between the *F_{st}* based detection of parallel candidate genes and the LFMM set. To me this is a real head scratcher!

7. The DMC analyses were only used for the small set of 61 genes identified via both *F_{st}* and LFMM, and then only with pairs of populations at a time that shared signal of adaptive evolution. Can this method be used to locate peaks of parallel adaptation using whole genome scans or is it really limited to individual genes? If the former, why only use it on the 61 genes? Finally how do the authors interpret the fact that only a minority of the 61 HIGHLY filtered candidate genes pass the DMC test of positive selection? This odd result was not discussed.

8. I really liked the findings of the clearly separate mutations TPC1 near the channel. But the paper did not really clarify for my why disrupting the ionic selectivity control of this channel might be adaptive in serpentine soils (or even if this has anything to do with Ca instead of Na) - - I realize it'd be pure speculation but a sentence or two added to this section would help me think a bit more about this gene's evolution at least.

9. Finally, as I am not sold by the framing of the paper around the idea of polyploidy and therefore the uniqueness of this study, I am not quite sure whether the findings, if substantiated more as outlined above, would be sufficiently novel for this journal. The authors cite several examples of parallel adaptation via shared variants and there are many more. But perhaps a rethink about the framing of the key findings will help.

Reviewer #3 (Remarks to the Author):

In this article, Konečná and collaborators investigate the mode of evolution of adaptive alleles in independent adaptations to serpentine soil in the autotetraploid species *Arabidopsis arenosa*. Using an elegant experimental design, the authors demonstrate that the parallelism between independent serpentine soil colonisation is mainly associated with the selection of shared segregating variants, with a few exceptions where selected alleles have evolved from de novo mutation. Polyploidy is thought to promote adaptation, but the underlying evolutionary mechanisms are still debated. This article suggests that a high level of functional standing variation in polyploidy facilitates adaptation to new environments. Furthermore, although the contribution of standing variation to rapid adaptation to new environments has been demonstrated in several species, empirical evidence of independent mutations at the same locus, especially in polyploidy lineages, are more limited. This study provides evidence that intense selective pressures can also promote parallel de novo mutation in autopolyploid species where theory predicts a reduced natural selection efficiency. Structural modelling of the proteins encoded by these alleles suggests a dominant behaviour of these new mutations in accordance with expectations. Indeed the increase in gene copy number in autopolyploid is expected to delay the

fixation of adaptive alleles and favour dominant mutations as a source of evolution. The analyses have been rigorously conducted, and the methods are described in great detail. While this study has great potential to make an important contribution to the field by revealing how recent whole-genome duplication may influence evolutionary mechanisms, the manuscript could still be improved.

General comments:

1-A main novelty in this study is the investigation of the contribution of different modes of evolution in autopolyploids. Theoretical work suggests that polyploidy promotes adaptation because polysomic masking favours the accumulation of functional genetic variants and the larger mutational target size increases the probability of adaptive de novo mutations. On the other hand, the increase in gene copy number is likely to slow down the fixation of new adaptive mutations. Testing these predictions is important because it addresses how individuals' genetic constitution may influence evolutionary dynamics and facilitate adaptation to new conditions. This study suggests that polyploids adapt to new conditions mostly through the fixation of standing variants. However, without a comparison with independent serpentine adaptations in a diploid lineage, it is not easy to assess the extent to which the 'ploidy' factor influences the mode of evolution. Perhaps, the authors could provide such a comparison by conducting similar analyses using available *A. lyrata* data (Turner et al., 2010) or shift the focus away from autopolyploidy's influence on the modes of adaptation.

2-As acknowledged by the authors (Line 423-427), this study is strongly biased towards the identification of standing variants as sources of beneficial mutations. The analysis focuses on loci showing sign of selection in at least two independent serpentine adaptation events and which are more likely to represent selection on shared variants. Assuming that the traits underlying serpentine adaptation are highly polygenic, the probability that independent de novo mutations hit the same target is low. It seems, therefore, tricky to estimate the contribution of standing variation over de novo mutations. I think it would still be informative to present the total number of loci contributing to serpentine adaptation in each population pairs and the proportion of parallel adaptive variation. Maybe the presence of serpentine specific variants among the 'non-parallel adaptive variants' could also be investigated as a rough estimate of the mode of evolution.

Minor comments

3- This study nicely illustrates the power of studying repeated evolution to highlight molecular features of key ecological importance. It suggests that independent de novo mutations at TPC1 contributed to parallel serpentine adaptation in *A. arenosa*, highlighting TCP1 as a putative evolutionary hotspot for serpentine adaptation. While structural homology models and allele frequencies strongly support this idea, the evidence is still only correlative. The mode of evolution is determined through a modelling approach that relies on a set of assumptions, and that is also probably quite sensitive to the quality of variant calling (see also comment 8 regarding the best fitting scenario). Without a functional study validating the modelling approach and demonstrating that the novel variants affect TPC1 function and facilitate adaptation to serpentine soil, the causative nature of these mutations is still speculative. Although this is not the study's main result, I think this should be acknowledged, and more generally, possible confounding factors (background selection, etc...) should be discussed.

4- I wonder to which extent the population pairs 3 and 4 should be kept as independent events since the scenario of their independent origin is not well supported. Interestingly, they also show a larger number of parallel differentiation candidates. Maybe, the authors could be more 'stringent' in their analyses and focus on clear independent events.

5- It seems appropriate to present pairwise F_{st} for all possible population comparisons in the supplementary information.

6- Please justify (in the method section) the choice of treating population pairs as random variables in your analyses. In relation to that, it may be interesting to perform a clustering analysis on the ionomic data to clearly illustrate the similarity between the different sites and confirm that these independent events reflect adaptation to similar soil types. The population pair 5 seems a bit different.

7- The motivation behind the investigation of divergence at transposable elements variations is not

well presented in the paper. I suppose it is related to the effect of polysomic masking on TE dynamics and the possible contribution to adaptive genetic variation. Perhaps, the authors could discuss the predictions tested in the introduction and their conclusions in the discussion. Also, I was wondering why the loci with variations in TE did not come out in the SNP – based scans for directional selection or Environmental association analysis. If I understand correctly, the TE analysis only considered region containing TE variations and, as a result, is much more 'permissive'. If this is true, what arguments justify using a different threshold and not consider the non-TE variants when selecting the 1 % outlier windows? And thus to include these regions for further analyses. Maybe the authors could indicate how these regions relate (which quantile) to the windows identified when considering all SNPs.

8- From the methods section or the manuscript itself, it was unclear to me how the authors assessed the significance of the difference in MCL for different models of convergent evolution (e.g., Based on figure 2e, selection on de novo mutation has only a slightly higher likelihood than selection on standing variation – how did the authors determine the significance of the difference in MCL?). Were the parallel models only compared to the neutral model or also compared to each other? Could the authors also clarify why they choose to position the selected site at 8 equally distributed location across the candidate genes? Wouldn't it make sense to set these positions based on LD decay?

9-Line263: "Assuming linkage between TE variant and surrounding SNPs" -could this be formally tested?

10-Line 311 'from the three serpentine populations'. Could the authors please clarify why they focused on 3 populations?

11-Line 372: "which have been shown to be important for stabilising Asn627 in *A.thaliana*". Please provide the corresponding reference.

12-Line 442: "⁶" should be removed or replaced

13-Line 603: "significantin", should be " significant in"

14- Line 773: Shouldn't "vcf" be "VCF"

Reviewer #4 (Remarks to the Author):

This is an excellent study looking into the genes in the tetraploid *A. arenosa* that are leading to adaptations to serpentine soils, typically rich in Mg and Ni compared to non serpentine sites used for comparisons. The authors look for signatures of selection on serpentine sites and predict that most comes from existing allelic variation which is selected independently with the colonisation of serpentine soils. This is an important finding for those interesting in polyploidy species establishment and for those interested in understanding how species become tolerant to serpentine soils.

The work is very tightly written, at times too tightly written, so indefinite article are sometimes missing for my taste, but more critically, explanations of exactly what the figures are showing is frequently minimal. No better example of this are the descriptions to Figure 2. This needs unpacking, and the legend and text appropriately expanded. The numbers on top of the Fig 2a bars means what, should they sum to 207? What do the bar colours mean (the legend being out of focus)? I think for 2e there is no reference at all about what we are to read from it.

This unpacking is true also of Figure 3. Do we need 'd' at all? The yellow sticks are not discussed in the text I don't think, we are told there are three colours, blue, marine and purple, I can resolve only two 'blue-based' colours. The text says "the Tyr side chain can adopt one of two broad conformations, either sticking into the channel, where it occludes the opening, or sticking away from the channel and directly into surrounding residues.....". When I compare the expanded boxes I struggle to see what is the point being made. The feature they highlight needs to be arrowed or something...so the reader can compare S and N (the expanded bits are not always being identical to the unexpanded source does not help).

There is also a necessity to better integrate the literature elsewhere, including from the authors

themselves (Konečná et al., 2020), to compare these findings with what has been seen and reviewed before with regard to other serpentine systems. This includes previous findings (including in *A. arenosa*) using QTLs that have identified TPC1 and other genes for Ca, K and Ni homeostasis.

Having asked for expansion of the text in all these areas, I remain confident that the work is overall excellent and a substantial contribution to our understanding of polyploid evolution and serpentine adaptations.

Additional points

- Fig. 1. (A) plant picture is very poor and not needed, ideally the map of Europe should be expanded. The complementary figure in Supplementary materials needs to be coordinate with towns, rivers or something similar.
- The text says "only the bioavailable concentration of Mg, Ni, Co, and Ca/Mg ratio consistently differentiated both soil types". This is not the case for Mg and it should be removed.
- The reference list appears twice.
- There is a spurious ⁶

Konečná V, Yant L, Kolář F. 2020. The Evolutionary Genomics of Serpentine Adaptation. *Frontiers in Plant Science* 11(2004).

EDITOR REMARKS

Thank you again for submitting your manuscript "Parallel adaptation in autopolyploid *Arabidopsis arenosa* is dominated by repeated recruitment of shared alleles" to Nature Communications. We have now received reports from 4 reviewers and, after careful consideration, we have decided to invite a major revision of the manuscript.

As you will see from the reports copied below, the reviewers raise important concerns. We find that these concerns limit the strength of the study, and therefore we ask you to address them with additional work. Without substantial revisions, we will be unlikely to send the paper back to review. In particular, we expect the problems of low number of sequenced plants from each population (Reviewer #2) and the independence of population pairs (Reviewer #2 and #3) could be fully addressed in the reversion.

If you feel that you are able to comprehensively address the reviewers' concerns, please provide a point-by-point response to these comments along with your revision. Please show all changes in the manuscript text file with track changes or colour highlighting. If you are unable to address specific reviewer requests or find any points invalid, please explain why in the point-by-point response.

REVIEWER COMMENTS

Reviewer #1 (Remarks to the Author):

This manuscript studied the genetic basis of parallel adaptation or evolution using five serpentine/non-serpentine populations pairs of autotetraploid *Arabidopsis arenosa*. Based on the selection scan and modeling analysis of the population genomics data, authors discovered substantial parallelism in candidate genes involved in ion homeostasis. In summary, this study suggested that polyploid populations can rapidly adapt to stress environment using both standing and novel genetic variation. Overall, this is a generally interesting study.

1) Figure 2a, apparently, no any differentiation gene shared across all the five pairs, is there any possible reason for this.

We consider the absence of differentiation candidates in all five population pairs to be a composite effect of a complex genetic architecture (demonstrated by a process-level convergence), potentially combined with partial (soft) sweeps that are particularly likely to occur in autotetraploids (Monnahan and Brandvain, 2020) yet their detection based on unphased short-read data is challenging. Indeed, the serial decreases in the total number of shared candidates from detection in any two to any three populations and then again from any three to any four populations is consistent with then observing no shared alleles across all five contrasts: the pattern is quite linear from one level to the next (Fig. 2a). This degree of (non-)parallelism is not uncommon in similar

systems (see also e.g. Bohutínská, Vlček, et al., 2021); yet in most of them it has not been measured in so many populations as in our study and thus we have the power to show where any parallelism falls off (Preite et al., 2019; Bohutínská, Alston, et al., 2021).

Regarding genetic architecture: the obvious polygenic basis of the adaptation is likely combined with the complexity of the pathways involved in the adaptation, causing changes in different genes from the same pathway (functional parallelism rather than genic). Illustrating this, we discovered significant overlap in functional categories (gene ontology terms; with max 4-fold overlap) that were inferred from outlier candidate lists from each population pair. To strengthen this, we present additional GO enrichment analysis (biological processes, cellular components, and molecular functions) in Dataset S4 and Fig. S7.

Regarding possible partial (soft) sweeps, a recent simulation-based study on autotetraploids revealed that particularly in loci with (partially) dominant alleles there are extended fixation times, which may cause some sweeps to remain undetected, especially in large populations (Monnahan and Brandvain, 2020). This, in combination with our conservative 99% threshold (to reduce false positives), and low genome-wide differentiation between populations, could leave some loci undetected. This is illustrated by the fact that using a more relaxed 97% F_{ST} -quantile threshold resulted in 4 genes overlapping in the fivefold contrast. We, however, prefer keeping the stringent F_{ST} selection criteria, as this minimizes false positive rates.

To reflect this we added the following to Results I. 235-242:

“The absence of common candidates across all five population pairs may reflect a complex genetic basis of the traits allowing for the modulation of the same pathway by different genes in some populations. This is supported by significant functional parallelism, i.e. higher than random number of overlapping GO terms that were repeatedly identified by separate enrichment analyses of outlier gene list from each population pair (Fig S7, Dataset S4h-j). Additionally, adaptation via partial (soft) sweeps, which are likely to occur in autotetraploids (Monnahan and Brandvain 2020), might have further limited the power of our divergence scans in some loci and populations.”

We also modified the Methods section I. 723-731:

*“We inferred potential functional consequences of the candidate gene lists using gene ontology (GO) enrichment tests within “biological processes”, “molecular functions”, and “cellular components” domains. We applied Fisher’s exact test ($p < 0.05$) with the “elim” algorithm implemented in topGO v.2.42 R package (Alexa and Rahnenfuhrer, 2020). We worked with *A. thaliana* orthologs of *A. lyrata* genes obtained using biomaRt (Durinck et al., 2009) and *A. thaliana* was also used as the background gene universe in all gene set enrichment analyses. The used “elim” algorithm traverses the GO hierarchy from the bottom to the top, discarding genes that have already been mapped to significant child terms while accounting for the total number of genes annotated in the GO term (Grossmann et al., 2007; Alexa and Rahnenfuhrer, 2020).”*

2) For Figure 2b, the GO enrichment was showed as molecular function, it is better to show biological processes as well.

Thank you for this suggestion. We now performed the GO enrichment for all three categories - biological processes, molecular functions, and cellular components. To clarify that in the text we added l. 222-227:

“However, in support of their relevance a gene ontology (GO) analysis of the 2245 candidates from all five population pairs shows significant enrichment (Fisher’s exact test; $p < 0.05$) of biological processes (BP), molecular functions (MF), and cellular components (CC) considered relevant to serpentine adaptation (Brady et al., 2005; O’Dell and Rajakaruna, 2011) such as inorganic anion transport, ion homeostasis, post-embryonic development, and calcium transmembrane transporter activity (Fig 2b; Dataset S4a-c).”

Further, we updated Fig. 2b with these new results and its legend l. 275-277:

“Gene ontology enrichment of the candidates (across all population pairs); GO categories: biological processes (BP) and molecular functions (MF); for complete list of GO terms see Dataset S4a-b.”

3) For TE-associated differentiation candidate genes, it is not clear to what extent these differentiated candidate genes overlapped with the differentiated genes based on SNP. It is not mentioned in the manuscript.

Thank you for this point. We clarified this aspect and added the following information to the Results, l. 300-301:

“In comparison with the list of candidates based on SNPs (for the same four population pairs, $n = 1853$), we observed the overlap of 46 genes.”

Please see also the answer to Reviewer #3 point 7 for further discussion.

Reviewer #2 (Remarks to the Author):

This paper claims to show, using genome sequence data from 78 plants from 10 populations, that independent adaptation to serpentine soils by different populations of the autotetraploid plant *Arabidopsis arenosa* involved "substantial parallelism." By this the authors mean that overlapping sets of candidate genes experienced positive selection in two or more of the five serpentine populations. The paper further claims that most instances of parallel adaptive evolution in candidate genes involved the selective recruitment same genetic variants across populations, rather than parallel adaptation via separate new mutations. However the authors do convincingly document a single case of separate new adaptive serpentine mutations in a TPC1 gene that appear, via protein modeling, to disrupt an ion channel.

The whole paper is framed around the idea that autopolyploid species may harbor more "standing" genetic variation than diploids because there are more gene copies per individual (4 vs 2), and so parallel adaptation via shared variants might be an especially prevalent mode of evolution in species like *A. arenosa*.

While I was initially quite interested and impressed by these claims and the framing of the paper, the closer I examined the underlying evidence the less convinced I was that the data and its analysis convincingly upholds the conclusions. Below I outline my main concerns.

1. I was not convinced that the framing of the paper, in the introductory and concluding paragraphs, around issues of diploid vs polyploid patterns of parallel adaptation was useful or particularly relevant to this study. First, is it really established that autotetraploids actually tend to have greater nucleotide diversity than their diploid progenitors? I could not find any such pattern among the papers cited by the authors. In fact the authors' earlier paper comparing diversity between diploid and polyploid *A. arenosa* found no difference in pairwise diversity and only very slight increase of non-synonymous segregating site diversity but no difference in synonymous diversity. As the authors pointed out, the young age of this autopolyploid might contribute to this lack of a pattern. And in fact the diversity levels for this autopolyploid given in Table 1 are right in line with those of many diploid outcrossing plant species. (question - are those values for π only for 4-fold degenerate sites? if so that should be stated clearly in the table)

We thank the reviewer for these important comments. We agree and have thus substantially modified the framing of the paper accordingly. Given the lack of diploid *A. arenosa* adapted to stressful serpentine conditions, we have made a clear shift in the framing to focus on the sources of parallel autotetraploid candidate allelic variation (below). The reviewer is also absolutely correct that there is only a very minor increase in diversity in autotetraploid *A. arenosa*, specifically at non-synonymous sites, according to our previous report (Monnahan et al., 2019), and it is not generally established that autotetraploids have larger nucleotide diversity as compared to diploids (due to a lack of empirical data). We have therefore modified our treatment of this, softening this general statement, but also demonstrating the slightly increased diversity in our serpentine populations in the current study (Table 1). We also modified our original potentially misleading formulations implying comparison of tetraploid and diploid populations to clarify this was not the aim of our study. For example, we removed the statement on "increased variation" from Abstract (l. 40-41) and modified parts of Introduction to highlight our focus on adaptation on understanding the sources of adaptive variation in autotetraploid populations (l. 109-112):

"Relative contributions of pre-existing vs de novo genomic variation to adaptation are poorly understood, especially in polyploid organisms."

*"As a similar comparison was not feasible for *A. arenosa* diploids, which have not been observed on serpentine soils, we refrain from direct inter-ploidy comparisons and instead*

specifically address the precise sources of adaptive variation in autotetraploid populations, a challenging and thus far unaddressed question.”

And Discussion, l. 448-453:

*“Our approach informs on natural sources of adaptive variation in autotetraploid populations, which has not been investigated genome-wide in a natural polyploid system. Yet we refrain from direct comparison with diploid ancestors because diploid serpentine populations are not known in *A. arenosa*, leaving a space for further study investigating other species encompassing multiple ploidies facing the same environmental challenge.”*

We also added the information that all values presented in Table 1 are based on fourfold-degenerate sites.

2. The authors collected ~8 plants from 5 pairs of adjacent serpentine and non-serpentine populations, and then for most of the analyses of Fst, LFMM, and DMC these 5 pairs were treated as if they were essentially independent of each other. However the PCA in Fig 1 clearly indicates that based on genome wide patterns of variation there really are only 3 main groupings of populations that are clearly divergent from each other (and the two populations in pair 3 clearly do not cluster together). The authors need to justify, on population genetic terms, their use of 5 pairs rather than 3 groups, and explain how this choice may influence their downstream conclusions about selection.

We are grateful for this important comment. We are afraid this concern may partly reflect a potentially confounding Principal component analysis (PCA) result originally presented in Fig. 1b. PCA can be misleading in complex multi-group hierarchical population structure (as in our case of 10 populations collected in multiple regions) because the reduction of dimensionality does not fairly depict relationships among all populations in a simple 2D plot. Indeed, the population pairs S1-N1 and S2-N2 clearly separate along the third axis (PC3) that explains nearly the same amount of variation as PC2 (3.6% and 4%, respectively). To clarify this, we now present additional axes combinations (PC1-PC3 and PC2-PC3) as new panes in Fig. S1b-d. As all these figures cannot be incorporated into Fig. 1 in the main manuscript, we replaced the original PCA plot by a Treemix allele frequency covariance graph (originally as Fig. S1b) which visualizes the relationships among all populations in a more balanced way and also shows the separate position of all five serpentine populations (please see the new Fig. 1b).

Importantly, to go beyond the descriptive approaches we further tested for independent colonisation of each serpentine barren by specifically designed coalescent simulations, iteratively over all possible population combinations. To do so, we simulated alternative scenarios of divergence and gene flow in fastsimcoal and selected preferred models based on Akaike Information Criteria (AIC). This approach is suitable for autotetraploid populations (Arnold et al., 2015) and was successfully applied to resolve complex population scenarios in recently divergent species and populations (Meier et al., 2017; Pfeifer et al., 2018; Monnahan et al., 2019). In all combinations of our populations, the scenario of independent colonisation from spatially adjacent

non-serpentine populations was preferred over the alternative clustering of populations by substrate type (please see Fig. S2 for distribution of AIC values and Dataset S2). For example, even despite population S3 approaches the N5-S5 cluster in the PCA, this population still represents a distinct, independent serpentine lineage more likely than being a sister of another serpentine population (Fig. S2). The intermediate position of Eastern Alpine population S3 in PCA thus likely reflects a complex pattern of sorting of ancestral polymorphism in this highly diverse glacial refugium of Alpine flora (Schönswetter et al., 2005). Such result of coalescent simulations, robust to particular combination of populations used, in our view justifies separate divergence-scan analyses of each of the five serpentine populations (i.e., independent instances of serpentine colonization), which also represent the most natural units for the population-genomic analyses (please see also our reply to Reviewer #2 point 4). To make this clear, we highlighted the results of fastsimcoal in the revised version by doing the following changes:

- (1) l. 132-142: we rephrased the text of Results to be more explicit on fastsimcoal model comparison and conclusions:

“We thus further tested the independent colonization of each serpentine site by coalescent simulations. Consistently over all possible pairwise iterations of S-N population pairs (n=10) the scenario of independent colonisation of each serpentine site was more likely than any scenario assuming sister position of two S populations (Fig 1c). Note that subsequent gene flow between substrate types within each S-N population pair was unlikely as the assumption of migration within each population pair had not significantly improved the model fit (Fig S2, Dataset S2). Reflecting the independent origin of the five serpentine populations, we analysed each serpentine colonisation event separately in the following analyses to take into account neutral population structure in the data, using the spatially closest N population as a contrast where needed.”

- (2) To further clarify the fastsimcoal results we prepared a new summary table (Dataset S2) with scenario selection results including the max likelihoods values, mean AIC values, and adjusted p values from Tukey multiple comparisons of mean tests for each population quartet and simulated scenario.

Finally, to address this concern from a different point of view, we also re-analysed our data from the perspective of how our choice of five population pairs influences the downstream analyses and main conclusions. We iteratively excluded one of the population pairs from the Eastern Alps region (pairs 3-5; which showed the most complex population structure) and re-calculated the levels of sharing of serpentine adaptation candidates (inferred by DMC) using the remaining four population pairs. By doing this we scrutinized the major result of our study, i.e. dominant role of shared variation in autopolyploid adaptation. As can be seen from the table below, all such iterations demonstrate that shared origin is the dominant source of variation (87-100 % of cases) also when only four population pairs are compared.

Inferred variation sources in the candidate cases of parallelism for originally 1% and 3% differentiation outliers and various iterations of population pairs from the Eastern Alps region. Note: The only candidate for de novo origin under the more stringent 1% threshold is TPC1.

	S1-N1, S2-N2, S3-N3, S5-N5		S1-N1, S2-N2, S3-N3, S4-N4		S1-N1, S2-N2, S4-N4, S5-N5	
	1% Fst outliers	3% Fst outliers	1% Fst outliers	3% Fst outliers	1% Fst outliers	3% Fst outliers
Shared origin	16	46	14	66	18	67
De novo origin	0	4	1	10	0	2

3. I am quite concerned about the low number of sequenced plants per population and how this might reduce statistical power and increase noise and false positives in the population genetic analyses of selection. Each plant has 4 copies of each site in the genome, and so there are a total of 5 possible genotypes at each bi-allelic SNP. It sounds like for each SNP these genotypes were called for each plant, and then population allele frequencies were estimated by essentially counting up the numbers of each allele in each of the 8 genotypes in each population - ideally each population allele frequency would be based on 32 total copies of each SNP. With a mean of 21x per site per individual, but with plenty of variation around this mean, how confident are we in the individual genotype calls and the resulting estimates of population allele frequencies given this variation of coverage, the 4 copy per site per individual issue, and the low number of individuals/allele counts per population? As far as I can tell all analyses take the allele frequencies estimates at face value, and they do not take into account all of the different sources of uncertainty.

We agree with these thoughtful points and definitely acknowledge in agreement that variant calling in autotetraploids is challenging, as well as practical limitations to resources for sampling. In summary, to test for these very helpful points, we now present comparisons of site frequency spectra (SFS) summarizing our allele frequency estimates (Fig. S15) as well as perform dedicated tests of how subsampling from a very highly sampled population (S3; n=17; Fig. S16) down to lower sample numbers affect population allele frequency estimates.

First, we just like to note that the sampling approach is in fact proved fruitful: we used our established approach benefiting from variant calling algorithms specifically designed for autopolyploids as implemented in GATK, which has been very fruitfully applied to a substantial literature of genome-wide investigations of *A. arenosa* (Hollister et al., 2012; Yant et al., 2013; Arnold et al., 2015, 2016; Baduel et al., 2018, 2019; Marburger et al., 2019; Monnahan et al., 2019; Bohutínská, Handrick, et al., 2021; Bohutínská, Vlček, et al., 2021).

However, we do not rely on the literature and do much appreciate the reviewer's thoughtful concerns. Therefore, to test whether variation in coverage per site or low sample size biased our results we conducted an analysis of SFS with samples from S3 population. We focused on this

population because additional 9 sequenced samples (without associated soil data thus not used further in our study) were available from the previous study (Arnold et al., 2016) and could therefore downsample from there to test for consistency of results using more or fewer sampled individuals. In total we had 17 individuals, thus 68 chromosomes sampled. To test that variation in depth of coverage does not bias the shape of site frequency spectra we randomly subsampled the pool of alleles to 32 (8 individuals) and we assessed variation in SFS in 10 subsampling rounds using a designated program *est-sfs* (Keightley and Jackson, 2018). This analysis is now presented as Fig. S16 and referred from Methods in the main text, l. 558-562:

*“Our site frequency estimates, which were constructed by program *est-sfs* (Keightley and Jackson 2018), are likely not biased by the number of individuals as was demonstrated by consistent site frequency estimates when subsampling one more deeply sampled population to the most common number of 32 chromosomes (S3; when including additional nine individuals Arnold et al., 2016) (Fig S16).”*

Fig. S16 Site frequency spectrum from 10 iterations of random subsampling to 8 individuals (i.e. 32 alleles) from all available 17 individuals of S3 population (including the data from Arnold et al., 2016). Y shows proportion of sites with an allele of given frequency X. Black bars show upper and lower 95 quantile over the replicates. Very narrow range of the estimates implies that putative sampling and coverage bias has negligible effect on the allele frequency estimates summarized in the site frequency spectrum.

We also determine whether our genotyping approach leads to biologically meaningful allele frequency estimates that are congruent with the previous study based on more representative range-wide sampling of *A. arenosa* (Monnahan et al., 2019). To clarify this, we now present SFS of all our populations as a new Fig. S15 (see the plot below for population S3 including all available individuals as an illustrative example). To assess whether our SFS (and thus overall allele frequency estimates) give biologically relevant results we moreover compared neutral SFS based on fourfold degenerate sites to SFS of sites under selection based on zerofold degenerate sites. In line with expectations (Boyko et al., 2008; Lawrie et al., 2013) and previous results in *A. arenosa* based on a larger population sample (Hollister et al., 2012; Monnahan et al., 2019) we observed a clear excess of singletons in zerofold compared to fourfold SFS reflecting the footprints of purifying selection.

We extended the Methods I. 556-558:

“This way of genotyping leads to allele frequency estimates which are well-comparable with previous estimates (Hollister et al., 2012) including a broad range-wide A. arenosa sampling (Monnahan et al. 2019) (the site frequency spectra are presented in Fig S15).”

*Site frequency spectra of fourfold and zerofold sites for S3 population. Figure shows excess of singletons in zerofold sites suggesting genome-wide purifying selection in line with previous analyses in *A. arenosa* (Hollister et al., 2012; Monnahan et al., 2019). Note: based on the level of missingness we downsampled the data to the average number of observed alleles per particular site (56 alleles: pop S3 all individuals; 32 alleles: pop S3 (only individuals from this study)); sites with fixed alleles are not shown.*

4. I found the section outlining the F_{ST} scans for each of the 5 pairs of populations to be a bit perplexing. I have several issues/questions. First, it would be helpful to see the Manhattan plots for individual SNPs in the supplemental figures as well as that for the windowed data - this would help the reader to see how noisy or clear the data look. Second, why window based on 1kb windows rather than on windows of fixed # of SNPs? And is that 1kb of genome length, or 1 kb of sites that are not masked? Because SNP density is expected to vary tremendously, windows based on physical length may "overly flatten" peaks if there are a lot of SNPs or they may not be included if they are in a conserved region with < 10 SNPs. There's no obvious best way to approach this, but it might be worth seeing what windows of 5, 10, 20, etc SNPs look like compared to the 1 kb windows. Finally, it seems worth looking at additional comparisons of F_{ST} for serpentine vs non serpentine for other groupings of populations other than just the five pairs. For example, it seems compelling based on the PCA to group populations into the 3 clusters and examine F_{ST} by habitat within each of those groups. And it certainly is worth investigating what happens to F_{ST} by habitat if all populations are combined (calculating allele frequencies for each habitat using all plants).

Thank you for these constructive ideas and questions. It is clear that the reviewer is thinking deeply about these issues, which have occupied us also for some time. In response to these points:

-We added Manhattan plots for individual SNPs to Fig. S8. We also modified the original 1kbp-window Manhattan plots by plotting only the serpentine adaptation candidates in green dots, not all outlier windows associated within these candidate genes, which could lead to misleading interpretation due to different numbers of highlighted items.

-We agree: it is normally a difficult and perhaps arbitrary choice to use SNP- or 'kb'-based windows. Our general philosophy is that if care is taken to avoid artefacts produced by minimum (or dispersed SNP) data, there is no different either way in most cases. However, here there was a clear choice we needed to make: as the primary objective of our F_{ST} scans was to assay for the emergence of the *same genomic loci differentiated* repeatedly across population pairs, our absolute priority was to ensure homogeneity in the approach across distinct population comparisons. SNP-based windows are highly influenced by missing data in each population pair, while the 1kbp windows we used do not shift their location based on coverage: they may be excluded due to low coverage, but they otherwise remain constant across all population pairs. To compare selection candidates across population pairs in a homogeneous way, we needed to work with identical windows.

To clarify our motivation for using window-based F_{ST} scans, we added this explanation to Methods I. 653-656:

"We used windows of fixed length and thus with homogeneous position on genome across all population pairs (in contrast to windows defined by number of SNPs and thus varying in exact position and length) to facilitate comparisons of selection candidates across distinct population pairs."

-To ensure windows of appropriate size were used with regard to the linkage, we consulted the empirical decay of genotypic correlations that scale with linkage disequilibrium as estimated in *A. arenosa* populations covering these same lineages (150-800 bp; Bohutínská, Vlček, et al., 2021). We now explain this in Methods at l. 651-653:

“The window size of 1 kbp was selected to properly account for the average genome-wide LD decay of genotypic correlations (150-800 bp) previously estimated in autotetraploid A. arenosa (Bohutínská, Vlček et al., 2021).”

-To account for possible biases caused by varying numbers of SNPs per window, we also checked whether F_{ST} values vary with an increasing number of SNPs per window and this analysis is now provided as Fig. S18. For all analyses already in the original version of the article, we removed uninformative windows with < 10 SNPs in which shortage of data may lead to spurious estimates of F_{ST} . When keeping only informative windows with ≥ 10 SNPs, the correlation between F_{ST} and number of SNPs per window was extremely weak (Spearman’s correlation coefficient ranging from 0.02 to 0.04 across population pairs). Therefore, we do not expect that the varying number of SNPs per informative window would strongly bias our F_{ST} inferences. To explain our data are not biased by the varying number of SNPs we present the new Fig. S18 (see the plot for population pair S1-N1 below as an illustrative example) demonstrating the relationship between the number of SNPs and F_{ST} and we added the following explanation to Methods, l. 646-648 and l. 656-659:

“We calculated pairwise F_{ST} (Weir and Cockerham, 1984) for non-overlapping 1 kbp windows along the genome with the minimum of 10 SNPs per window to exclude potential biases in F_{ST} estimation caused by low-informative windows (Beissinger et al., 2015).”

“Our F_{ST} estimates are unlikely to be strongly affected by varying numbers of SNPs per window, as the correlation between F_{ST} and number of SNPs per window was very weak (Spearman’s rank correlation coefficient varied from 0.02 to 0.04 across population pairs; Fig S18).”

Relationship between F_{ST} estimates for each 1kbp window and the number of SNPs per window for population pair 1. Note: low-informative windows with < 10 SNPs were excluded; ρ = Spearman's rank correlation coefficient; the regression line is in blue.

We also freely acknowledge in the manuscript that our initial list of F_{ST} candidates is permissive (stated in Results, l. 217-220, already in the original version) and we further highlighted this in the revised version l. 220-227:

“These most inclusive lists must be interpreted with caution, as they are based on a simple assumption that the most differentiated regions are under directional selection (Holsinger and Weir, 2009). However, in support of their relevance a gene ontology (GO) analysis of the 2245 candidates from all five population pairs shows significant enrichment (Fisher's exact test; $p < 0.05$) of biological processes (BP), molecular functions (MF), and cellular components (CC) considered relevant to serpentine adaptation (Brady et al., 2005; O'Dell and Rajakaruna, 2011) such as inorganic anion transport, ion homeostasis, post-embryonic development, and calcium transmembrane transporter activity (Fig 2b; Dataset S4a-c).”

-Crucially, we do not interpret the results solely based on the F_{ST} outliers alone. Our evaluation of the relative role of shared vs. *de novo* candidates is strictly based on the candidates which were further validated by two orthogonal approaches: (i) environmental association analysis — a statistical approach leveraging the selective environmental factors of interest (individual soil elemental concentrations), (ii) likelihood-based model comparison in the DMC framework which allowed to discriminate parallel selection signals in S populations from both neutral scenario and alternative signals of selection sweeps also in N populations. This is explained in Methods l. 692-695:

“We made a final shortlist of serpentine adaptation candidates by overlapping the LFMM candidates, reflecting significant association with important soil elements, with the

previously identified parallel differentiation candidates, mirroring regions of excessive differentiation repeatedly found across parallel population pairs.”

and in Results I. 250-258 after some modifications:

“Finally, we overlapped the ‘LFMM candidates’ with the parallel differentiation candidates to produce a final refined list of 61 ‘serpentine adaptation candidates’ (Fig 2c, Fig S8, Dataset S6, Dataset S7a). This conservative approach aims to identify the strongest candidates underlying serpentine adaptation for further model-based inference of sources of variation in the next section. We note that this approach discards population-specific (private) candidates and cases of distinct genetic architecture of a trait (e.g. distinct genes affecting the same pathway) and thus cannot quantify the overall genome proportion which evolves in parallel. Importantly, however, it also minimizes false positives from population-specific selection and genetic drift.”

Finally, due to a clear structure in our data (populations/population pairs) we intentionally strictly avoid lumping populations into larger units, except for methods that can directly account for underlying population structure (LFMM).

(1) Such ad hoc lumping may lead to serious biases in F_{ST} estimates such as Wahlund effect (reduction in expected diversity due to merging of differentiated populations). We explained why we had not merged populations in Methods I. 641-646:

“Reflecting hierarchical structure in the data, we avoided merging multiple populations into larger units for the estimation of F_{ST} and strictly worked in a pairwise design. We admit that population S3 occupies a somewhat separate position and its ancestral non-serpentine population might have thus remained unsampled (or got extinct) - we therefore used the spatially closest population N3 as the most representative paired population available in our sampling.”

(2) Also the modelling approach — DMC, is based on an assumption of populations whose neutral variation is close to equilibrium, which would have been seriously biased in case of population merging (Lee and Coop, 2017).

5. Continuing with the F_{ST} based scans, I found the evidence for parallel adaptation at candidate genes to be weak and outlined in the paper in a somewhat misleading way. Comparing the peaks in the plots of windowed F_{ST} across the genome for each pair, it is immediately obvious that very few clear peaks are shared across populations. This may very well be due to low power or windowing issues to some extent but lets assume that its not for the time being. The authors find about 500 candidate genes have F_{ST} window values in the top 1% in each population pair. Amazingly if you ask whether the 500 in any particular pair are also found in another pair, only about 20-40 are shared, meaning that > 900 candidate genes found in just one of the 2 pairs. The situation is much less compelling if you pick 3 or 4 or 5 of the 5 pairs and ask how many of the candidate genes show high F_{ST} in those populations - just a couple of genes are shared in the 3 or 4 pairs and NONE

of the 2,245 candidate genes show high F_{ST} in all 5 populations. Taken at face value this is a remarkably low level of parallelism - the overwhelming pattern is of unique adaptations in each serpentine populations. This point is not clearly conveyed in the paper. Now, do we really believe in that exceedingly low level of parallelism? I strongly suspect that the small sample size/high noise/low power issue raised above causes both false positives (perhaps many of the ~500 candidate genes per pair are not actually under divergent selection but by chance F_{ST} is estimated to be high) and false negatives (noise leads to underestimates of F_{ST} in regions that are actually experiencing divergent selection).

We agree with the reviewer that the evidence indicates a very low degree of parallelism. This is actually consistent with an array of other studies (Lai et al., 2019; Preite et al., 2019; James et al., 2020; Ji et al., 2020; Bohutínská, Vlček, et al., 2021) and here we have the distinction of 5 separate contrasts to illustrate this. We appreciate the suggestion to reframe this for clarity which we have done; see also comments to Reviewer #1 point 1.

Indeed, we are also aware of possibly high number of false positive candidates resulting from relying solely on F_{ST} scans, which further lead to the decrease in the percentage of parallels (ranging from 2 to 4%). However, we note that even the observed overlap in 21 genes (the lowest overlap for any particular pair) is significantly higher than expected by chance (excess parallelism), as was shown also for almost all possible overlaps between two, three, and four population pairs (Fig. 2a). Considering the stochastic allele fluctuations in relatively recently diverged populations, even this low percentage of shared candidates is significant parallelism as any parallel hit by selection pointing towards shared genomic basis is rather unexpected.

That said, being fully aware of the limits of F_{ST} scans, we refrain from any interpretations based solely on those candidates and instead, we use them as initial starting lists that were further validated by complementary analyses such as environmental association analysis and DMC modelling leading to the short list of refined candidates. We specify this newly at various parts of Methods and Results I. 214-217 and I. 220-222 (please see also our response to your point 4):

“...we combined divergence scans and environmental association analysis to refine the list of loci for parallel selection modelling only to the candidates which repeatedly differentiated across multiple population pairs and were significantly associated with the selective soil environment.”

“These most inclusive lists <of F_{ST} outliers> must be interpreted with caution, as they are based on a simple assumption that the most differentiated regions are under directional selection (Holsinger and Weir 2009).”

Further, we agree with the reviewer that the lack of five-fold candidates may reflect complex genetic architectures of serpentine-relevant adaptive traits and thus the demonstrated functional parallelism (different loci from the same pathway may underlie the same trait), rather than the stochastic effect on gene parallelism. This demonstrated functional parallelism itself argues very strongly indeed against broadly artifactual results in the present study. Further, possible combination of complex genetic architecture with the presence of soft sweeps in polyploids is now

discussed in Results I. 225-242 (for further details, please see also our response to Reviewer #1 point 1).

“The absence of common candidates across all five population pairs may reflect a complex genetic basis of the traits allowing for the modulation of the same pathway by different genes in some populations. This is supported by significant functional parallelism, i.e. higher than random number of overlapping GO terms that were repeatedly identified by separate enrichment analyses of outlier gene list from each population pair (Fig S7, Dataset S4h-j). Additionally, adaptation via partial (soft) sweeps, which are likely to occur in autotetraploids (Monnahan and Brandvain 2020), might have further limited the power of our divergence scans in some loci and populations.”

Finally, we would like to highlight that our manuscript is ****not focused on quantifying parallelism**** in the sense of comprehensively quantifying the fraction of the genome that is evolving in parallel. Instead, we only leverage cases of clearly detectable parallelism based on conservative methods (presumably therefore a subset of all cases) to infer a set of the most reliable selection candidates in autotetraploid *A. arenosa*. To minimise potential confusion, we now even toned down the parallel evolution aspect in the manuscript title and we instead highlight the focus on a particular selective factor (serpentine):

*“Serpentine adaptation in autopolyploid *Arabidopsis arenosa* is dominated by repeated recruitment of shared alleles”*

We also removed the vague and potentially misleading term “substantial parallelism” from throughout the text.

In addition, we also explained this at the relevant part of the Results, I. 255-258:

“We note that this approach discards population-specific (private) candidates and cases of distinct genetic architecture of a trait (e.g. distinct genes affecting the same pathway) and thus cannot quantify the overall genome proportion which evolves in parallel. Importantly, however, it also minimizes false positives from population-specific selection and genetic drift.”

6. I do not fully understand the LFMM 2 analyses and whether this approach is a powerful method for identifying parallel adaptation or not. The paper does not clearly state that the unit of observation for this analysis is the individual and not the population, but because each of the 78 plants has genotype data at each SNP and results of chemical analyses of the soil around its roots during collection in the field, the authors attempt to use LFMM 2 to find associations between alleles at SNPs and soil data, while attempting to control or model into the analysis associations that might be caused by population structure alone. The problem is that the population structure in this collection of 10 populations is complex and hierarchical, and very clearly not simply 5 separate/independent pairs of populations (based on PCA and additional measures of structure in Supp. figs). And yet the latent factors used in this analysis to somehow model this structure is simply the 5 population

pairs. To what extent would the same number and identify of genes be identified if other latent factors were used that perhaps more accurately account for structure - for example what about using instead those 3 broad clusters in PCA space? The authors need to clearly explain how these issues of population structure may or may not have influenced the results. In addition it was not clear to me whether this method should in principle only identify broadly parallel adaptations, since I imagine that it would not flag a gene if it was only divergent in one of the serpentine populations? And yet there is again remarkably little overlap between the Fst based detection of parallel candidate genes and the LFMM set. To me this is a real head scratcher!

We appreciate these comments based on which we clarified the motivation for and setting of the LFMM approach. We highlight here that the aim of LFMM is not exhaustive detection of parallel adaptation but identifying a set of SNPs and associated genes that show consistent allele frequency variation in response to particular soil parameters associated with serpentine and thus further refinement of reliable serpentine adaptation candidates.

First, we specified that the unit of observation was individual, Methods I. 673-676:

“We tested the association of allele frequencies at each SNP for each individual with associated soil concentration of the key elements differentiating serpentine and non-serpentine soils: Ca/Mg ratio, and bioavailable soil concentrations of Co, Mg, and Ni.”

And also in the Results I. 245-247:

“This analysis quantitatively determines the association between each soil elemental concentration and SNPs across the genome in both S and N populations at the level of individual plants (in total 78).”

Second, following the reviewer's advice we also calculated the LFMM analysis using the suggested value of K=3. Running LFMM with three and then with five latent factors lead to negligible difference in the set of serpentine adaptation candidates. Specifically, the analysis assuming three latent factors resulted in a somewhat higher total number of LFMM candidates (K=3: 3447 as compared to K=5:2809) yet similar total number of serpentine adaptation candidates (K=3 -> 59 as compared to K=5 -> 61 with overlap in 55 genes). To clarify this, we added the following explanation of our choice of the number of latent factors to the Methods, I. 679-687:

“LFMM accounts for a discrete number of ancestral population groups as latent factors. We used five latent factors reflecting the number of population pairs. Due to hierarchical structure in the data (PCA based on ~1 M 4dg SNPs indicated five main components, yet the first three axes alone explained considerable variation; Fig S20), we also performed the additional analysis assuming three latent factors. As such analysis had only a minor effect on the total number of serpentine adaptation candidates (reducing their number by only two) we further used a candidate list based on five latent factors which corresponds

to the total number of population pairs, thus it is also directly comparable with the parallel differentiation candidates.”

Regarding the motivation, we highlight that the main aim of this approach was to refine the candidates in Results, l. 243-245:

“As a complementary approach, we inferred candidates directly associated with the distinctive chemical characteristics of serpentine soil by performing environmental association analysis using latent factor mixed models (LFMM) (Caye et al., 2019).”

We fully agree (and stated already in the original text, now at l. 255-257) that our approach could indeed underestimate the number of site-specific selection candidates. On the other hand, we believe that this shall not be a serious problem for our interpretation that is not focused on quantifying overall parallelism (proportion of parallel vs. site-specific, i.e. non-parallel candidates), but rather on inferring the sources of variation in the shortlist of reliable parallel candidates.

This is now clarified in l. 252-254:

“This conservative approach aims to identify the strongest candidates underlying serpentine adaptation for further model-based inference of sources of variation...”

As a result of this choice, the overlap of LFMM and F_{ST} candidates is naturally limited, possibly as a direct consequence that we were searching for the effect of the only subset of potential factors discriminating S and N populations (in order to efficiently filter out false positives, specified on l. 250): Ca/Mg, concentrations of Mg, Ni, and Co. Thus, the candidates inferred by F_{ST} scans likely also involve genes under selection due to the alternative factors differentiating S and N sites, like water availability and competition with other plants. On the other hand, the high number of LFMM candidates may reflect the varying power of this SNP-based approach and we thus prefer our conservative strategy focusing only on an overlap of LFMM with parallel divergence scans. As additional validation of our stringent candidate selection, there is strong GO enrichment of the candidates found by both F_{ST} scans and LFMM (termed ‘serpentine adaptation candidates’) for relevant functions such as regulation of ion transmembrane transport, voltage-gated calcium channel activity or voltage-gated potassium channel activity (specified in Results at l. 259-261 and Dataset S7b-d). We note, finally, that low overlap between the two methods can also be partly explained by the fact that the F_{ST} window-based approach requires homogenisation of signal from at least 10 adjacent SNPs, while LFMM works on a single SNP basis.

7. The DMC analyses were only used for the small set of 61 genes identified via both F_{ST} and LFMM, and then only with pairs of populations at a time that shared signal of adaptive evolution. Can this method be used to locate peaks of parallel adaptation using whole genome scans or is it really limited to individual genes? If the former, why only use it on the 61 genes? Finally how do the authors interpret the fact that only a minority of the 61 HIGHLY filtered candidate genes pass the DMC test of positive selection? This odd result was not discussed.

The DMC modelling approach was designed for inference of sources of variation in pre-selected genes not for genome-wide scans (Lee and Coop, 2017) and so far has been used exclusively to infer the evolutionary origin of the single gene or at most a handful of genes (Lee and Coop, 2017; Oziolor et al., 2019; Van Etten et al., 2020). To allow the application of this approach to multiple genes and population combinations, we were forced to pre-select genes from the genome-wide distribution using the above-mentioned approach. We admit that such motivation was insufficiently explained in the previous version and we clarify it by adding the following explanation (l. 214-217):

“...we combined divergence scans and environmental association analysis to refine the list of loci for parallel selection modelling only to the candidates which repeatedly differentiated across multiple population pairs and were significantly associated with the selective soil environment.”

Please also note the DMC model-based approach specifically tested for the modes of parallel selection identified based on the effect of hitchhiking on linked diversity within and among populations. Aside from the quite plausible validity of the low convergence (as discussed extensively above, and which is consistent with other studies also cited above), we can speculate that the low proportion of the pre-selected serpentine adaptation candidates passing the DMC threshold for *parallel selection* may reflect that during the DMC analysis: (i) we filtered out cases in which parallel selection was affecting not only S, but also N populations; or (ii) we used very stringent criteria for detection of parallel selection.

Regarding (i) we agree, this was not sufficiently explained and we thus modified Results, l. 322-329:

“This analysis indicated that parallel selection exceeded the neutral model for 62 out of the total 84 candidate cases of parallelism (i.e. cases when two population pairs shared one of the 61 serpentine adaptation candidates). To focus only on well-justified candidates of adaptation within the serpentine populations, we excluded additional 33 cases where the scenario of parallel selection with the highest MCL estimate in serpentine populations was not considerably higher (>10%) than this estimate in non-serpentine populations, which resulted in 29 candidate cases of serpentine adaptation parallelism.”

Regarding (ii) specifically, we required the difference in MCL (the maximum composite log-likelihood) between neutral and parallel selection scenario to be > 21 (the maximum of the distribution of the differences between neutral and parallel selection scenarios from the simulated data in *A. arenosa*; Bohutínská, Vlček, et al., 2021) and only parallel selection scenario with at least 10% difference in MCL estimate in serpentine compared to non-serpentine populations was taken into account (this is explained in Methods l. 762-770 and currently also in Results l. 325-329). Indeed, applying lower thresholds (minimum difference in MCL estimate set to 95% quantile of the distribution, i.e. the value of 3 and at least 3% difference in MCL estimate between S-N populations) increased the number of identified cases of parallelism to 38 (29 genes). Furthermore, considering this relaxed threshold for MCL difference between parallel and neutral scenarios (95% quantile) and keeping all genes regardless of the signal for selection to be

stronger in the S vs. N population led to 77 cases of parallelism (56 genes). For the main ms. we however prefer keeping the more conservative criteria and focusing only on well-justified candidates. To explain our conservative approach is not designed to infer a complete set of potentially adaptive loci (and thus many more false positives), but rather shortlist the top candidates, we also added the following to l. 343-347:

*“Note that our conservative approach, focused on identifying regions of repeated excessive differentiation and significant soil-related allele frequency differences is not designed to cover the entire range of adaptive loci. Further research is thus needed to comprehensively cover the complete landscape of adaptation in autotetraploid *A. arenosa*.”*

8. I really liked the findings of the clearly separate mutations TPC1 near the channel. But the paper did not really clarify for my why disrupting the ionic selectivity control of this channel might be adaptive in serpentine soils (or even if this has anything to do with Ca instead of Na) - - I realize it'd be pure speculation but a sentence or two added to this section would help me think a bit more about this gene's evolution at least.

Thank you; we found those findings exciting too. We agree some elaboration would be helpful and we therefore add (l. 434-439):

“Dedicated, electrophysiological single vacuole conductance experiments are required to explore functional changes in detail at TPC1. However, the exceptionally suggestive convergent changes we discovered and modelled to structures at the pore selectivity gate force the speculation that they mediate change in the relative conductance of the divalent cations Ca^{2+} and Mg^{2+} , the highly skewed ratios of which stand as hallmarks of serpentine soils (O'Dell and Rajakaruna, 2011).”

9. Finally, as I am not sold by the framing of the paper around the idea of polyploidy and therefore the uniqueness of this study, I am not quite sure whether the findings, if substantiated more as outlined above, would be sufficiently novel for this journal. The authors cite several examples of parallel adaptation via shared variants and there are many more. But perhaps a rethink about the framing of the key findings will help.

Thank you for your frank comments. It is helpful so we can reconsider the framing of our study and how it is most informative. We see the main novelty of this study to be understanding the contribution of different modes of evolution in autopolyploids: theoretical work suggests that polyploidy should promote adaptation because polysomic masking favours the accumulation of functional genetic variants and the larger mutational target size increases the probability of adaptive *de novo* mutations. On the other hand, the increase in gene copy number is likely to slow down the fixation of new adaptive mutations. Our data makes an early empirical assessment relating to these predictions. Therefore, our study is important because it addresses how an individual's genetic constitution may influence evolutionary dynamics and facilitate adaptation to new conditions.

Following this and the above comments, we included additional publications focused on parallel adaptation from standing vs. *de novo* variation in diploid systems to illustrate the context. We do still feel this study of the sources of autotetraploid variation is highly valuable as, to our knowledge, our study represents a first empirical inquiry of repeated adaptation in a natural autopolyploid system including the investigation of the contribution of different evolution modes (please see also our reply to Reviewer #2 point 1). Below, we highlight how we believe the novelty of our study is now appropriately highlighted with respect to polyploid adaptation:

Introduction, l. 59-62:

“While WGD is clearly associated with environmental change or stress (Peer et al., 2017; Peer et al., 2020), the precise impact of WGD on adaptability is largely unknown in multicellular organisms, and there is virtually no work assessing the evolutionary sources of adaptive genetic variation in young polyploids.”

Discussion, l. 468-474:

“Such a dominant role of pre-existing variation is in line with the studies of parallel adaptation in diploid systems such as Littorina snails (Ravinet et al., 2016; Morales et al., 2019), stickleback fishes (Colosimo et al., 2005; Jones et al., 2012), Heliconius butterflies (Pardo-Diaz et al., 2012), Sinusuthora webbiana vinous-throated parrotbill birds (Lai et al., 2019), Ipomoea purpurea morning glories (Van Etten et al., 2020), Apis cerana Asian honeybees (Ji et al., 2020), or Coilia nasus fishes (Zong et al., 2020). Yet examples are lacking from autopolyploid systems, where a large pool of standing variation is expected by theory due to larger effective population size and polysomic masking of allelic variation (Peer et al., 2017; Baduet et al., 2018).”

Reviewer #3 (Remarks to the Author):

In this article, Konečná and collaborators investigate the mode of evolution of adaptive alleles in independent adaptations to serpentine soil in the autotetraploid species Arabidopsis arenosa. Using an elegant experimental design, the authors demonstrate that the parallelism between independent serpentine soil colonisation is mainly associated with the selection of shared segregating variants, with a few exceptions where selected alleles have evolved from *de novo* mutation. Polyploidy is thought to promote adaptation, but the underlying evolutionary mechanisms are still debated. This article suggests that a high level of functional standing variation in polyploidy facilitates adaptation to new environments. Furthermore, although the contribution of standing variation to rapid adaptation to new environments has been demonstrated in several species, empirical evidence of independent mutations at the same locus, especially in polyploidy lineages, are more limited. This study provides evidence that intense selective pressures can also promote parallel *de novo* mutation in autopolyploid species where theory predicts a reduced natural selection efficiency. Structural modelling of the proteins encoded by these alleles suggests a dominant behaviour of these new mutations in accordance with

expectations. Indeed the increase in gene copy number in autopolyploid is expected to delay the fixation of adaptive alleles and favour dominant mutations as a source of evolution. The analyses have been rigorously conducted, and the methods are described in great detail. While this study has great potential to make an important contribution to the field by revealing how recent whole-genome duplication may influence evolutionary mechanisms, the manuscript could still be improved.

We would like to thank the Reviewer for a precise summary of our manuscript.

General comments:

1-A main novelty in this study is the investigation of the contribution of different modes of evolution in autopolyploids. Theoretical work suggests that polyploidy promotes adaptation because polysomic masking favours the accumulation of functional genetic variants and the larger mutational target size increases the probability of adaptive de novo mutations. On the other hand, the increase in gene copy number is likely to slow down the fixation of new adaptive mutations. Testing these predictions is important because it addresses how individuals' genetic constitution may influence evolutionary dynamics and facilitate adaptation to new conditions. This study suggests that polyploids adapt to new conditions mostly through the fixation of standing variants. However, without a comparison with independent serpentine adaptations in a diploid lineage, it is not easy to assess the extent to which the 'ploidy' factor influences the mode of evolution. Perhaps, the authors could provide such a comparison by conducting similar analyses using available *A. lyrata* data (Turner et al., 2010) or shift the focus away from autopolyploidy's influence on the modes of adaptation.

We improved our explanation of the novelty of the ms in regards to tetraploid adaptation (please see our response to Reviewer #2 points 1 and 9). Also, we made sure that our study presents evolutionary sources of adaptation in autotetraploid populations, NOT inference the role of ploidy *per se* by comparison with the diploid ancestors (which unfortunately do not occupy serpentine stands). This is highlighted in Introduction (l. 109-112):

*“As a similar comparison was not feasible for *A. arenosa* diploids, which have not been observed on serpentine soils, we refrain from direct inter-ploidy comparisons and instead specifically address the precise sources of adaptive variation in autotetraploid populations, a challenging and thus far unaddressed question.”*

And Discussion, l. 448-453:

*“Our approach informs on natural sources of adaptive variation in autotetraploid populations, which has not been investigated genome-wide in a natural polyploid system. Yet we refrain from direct comparison with diploid ancestors because diploid serpentine populations are not known in *A. arenosa*, leaving a space for further study investigating other species encompassing multiple ploidies facing the same environmental challenge.”*

Although we appreciate the suggestion of using the *A. lyrata* example published by Turner et al., 2010 as a diploid contrast, we are afraid such study is not suitable for any reliable comparison with our data. Unfortunately, the study, performed already in 2009 does not meet the current standards for data presentation and sharing, specifically (i) they do not contain individual soil parameter data (environmental association analysis is not feasible), (ii) the populations grow in different environmental context (treeless boreal/subarctic habitats thus shaped by distinct selective pressures), (iii) crucially, the Turner et al data are based on pooled sequencing and, most frustratingly, (iv) the raw data are not available (from the manuscript itself, any metadata or supplemental data, nor by contacting the authors directly, which co-author Levi Yant attempted for this purpose: T Turner sent a short Excel list of 96 SNP outliers and S Nuzhdin did not reply to enquiries), making any re-analysis and thus direct comparison with our results impossible.

2-As acknowledged by the authors (Line 423-427), this study is strongly biased towards the identification of standing variants as sources of beneficial mutations. The analysis focuses on loci showing sign of selection in at least two independent serpentine adaptation events and which are more likely to represent selection on shared variants. Assuming that the traits underlying serpentine adaptation are highly polygenic, the probability that independent de novo mutations hit the same target is low. It seems, therefore, tricky to estimate the contribution of standing variation over de novo mutations. I think it would still be informative to present the total number of loci contributing to serpentine adaptation in each population pairs and the proportion of parallel adaptive variation. Maybe the presence of serpentine specific variants among the ‘non-parallel adaptive variants’ could also be investigated as a rough estimate of the mode of evolution.

For the proportion of shared differentiation candidates among populations please see Fig. 2a. Please see also Reviewer #2 point 5 for further discussion about the shared candidates.

We agree that the probability of hitting the same target by independent mutations is low, but it is also low for allele re-use in the same gene across populations. Overall, our manuscript is not focused on quantifying parallelism in the sense of identifying which fraction of the genome is evolving in parallel as is now specified in Results at l. 255-257. Instead, we only leverage cases of clearly detectable parallelism at both gene and functional level (presumably just a subset of all cases) to infer a set of reliable selection candidates and to assess the proportion of *de novo* mutations vs. pre-existing variation in these candidates in autotetraploid *A. arenosa*.

Minor comments

3- This study nicely illustrates the power of studying repeated evolution to highlight molecular features of key ecological importance. It suggests that independent de novo mutations at TPC1 contributed to parallel serpentine adaptation in *A. arenosa*, highlighting TCP1 as a putative evolutionary hotspot for serpentine adaptation. While structural homology models and allele frequencies strongly support this idea, the evidence is still only correlative. The mode of evolution is determined through a modelling approach that relies on a set of assumptions, and that is also probably quite sensitive to the quality of

variant calling (see also comment 8 regarding the best fitting scenario). Without a functional study validating the modelling approach and demonstrating that the novel variants affect TPC1 function and facilitate adaptation to serpentine soil, the causative nature of these mutations is still speculative. Although this is not the study's main result, I think this should be acknowledged, and more generally, possible confounding factors (background selection, etc...) should be discussed.

We thank the reviewer for this comment. We agree that this tantalising finding is not fully fleshed out, nor is it the present study's main result. However, we do indeed acknowledge this (l. 434-439):

“Dedicated, electrophysiological single vacuole conductance experiments are required to explore functional changes in detail at TPC1. However, the exceptionally suggestive convergent changes we discovered and modelled to structures at the pore selectivity gate force the speculation that they mediate change in the relative conductance of the divalent cations Ca^{2+} and Mg^{2+} , the highly skewed ratios of which stand as hallmarks of serpentine soils (O'Dell and Rajakaruna, 2011).”

Additionally, the homology modelling is unlikely to be affected heavily by variant calling because it only uses alleles at > 0.5 frequency.

4- I wonder to which extent the population pairs 3 and 4 should be kept as independent events since the scenario of their independent origin is not well supported. Interestingly, they also show a larger number of parallel differentiation candidates. Maybe, the authors could be more 'stringent' in their analyses and focus on clear independent events.

Thanks for this comment. We now provide an additional explanation indicating the independent origin of each serpentine population is more likely than their shared origin by a combination of allele frequency covariance analysis, Bayesian structure analyses and explicit demographic modelling in a coalescent framework. In addition, we also demonstrate that our main conclusions regarding the dominant role of shared variation do not change if we apply a more stringent approach and one of these population pairs is excluded. For further details, please see our answers to Reviewer #2 point 2.

5- It seems appropriate to present pairwise F_{st} for all possible population comparisons in the supplementary information.

Yes, we agree. Accordingly, we now add this information as a new Table S8.

6- Please justify (in the method section) the choice of treating population pairs as random variables in your analyses. In relation to that, it may be interesting to perform a clustering analysis on the iononic data to clearly illustrate the similarity between the different sites and confirm that these independent events reflect adaptation to similar soil types. The population pair 5 seems a bit different.

Good idea. We now add this explanation to Methods I. 799-801:

"To account for lineage-specific differences between population pairs, which are uninformative for the overall assessment of the fitness response towards serpentine, we treated population pairs as a random variable."

We now include a new Fig. S14: a heatmap of population clustering based on particular soil elemental concentrations. The distinctness of the pop pair S5-N5 reflects primarily high Ca content in the N5 population, which is not against the major trend in discriminating between the S and N populations, i.e. lower Ca/Mg ratio in S populations. We also added to I. 511-514:

"Although we observed a considerable variation in the overall soil chemistry in our samples (Fig S14) the principal soil factors differentiating between S and N populations were always the same – higher Mg, Ni, Co, and lower Ca/Mg in S populations."

Fig. S14 Heatmap of population clustering (UPGMA) based on particular soil elemental concentrations. Note: the elemental concentrations were centred and scaled using heatmap function in R scale="column" (for the original values see Table S4).

7- The motivation behind the investigation of divergence at transposable elements variations is not well presented in the paper. I suppose it is related to the effect of polysomic masking on TE dynamics and the possible contribution to adaptive genetic variation. Perhaps, the authors could discuss the predictions tested in the introduction and their conclusions in the discussion. Also, I was wondering why the loci with variations in TE did not come out in the SNP – based scans for directional selection or Environmental association analysis. If I understand correctly, the TE analysis only considered region containing TE variations and, as a result, is much more ‘permissive’. If this is true, what arguments justify using a different threshold and not consider the non-TE variants when selecting the 1 % outlier windows? And thus to include these regions for further analyses. Maybe the authors could indicate how these regions relate (which quantile) to the windows identified when considering all SNPs.

Thank you for this. We now add the motivation to the Results I. 292-294:

“Specific transposable elements (TE) families can be activated by abiotic stresses and possibly contribute to adaptation to challenging environments (Grandbastien et al., 2005; Baduel et al., 2019; Rogivue et al., 2019; Wos et al., 2021).”

We did only divergence scans to find TE-associated candidates as is described in Results I. 298-300:

“...we applied a similar differentiation outlier window-based workflow as specified above and identified 92–115 TE-associated candidate genes per S-N contrasts (Dataset S9).”

Indeed, we were specifically interested in the differentiated regions containing TE variants (and their putative functions inferred by GO enrichment analysis), Methods I. 712-714:

“...we calculated pairwise F_{ST} (Weir and Cockerham, 1984) using SNP frequencies (in the same way as specified above) for non-overlapping 1 kbp windows containing TE variant(s) for each population pair.”

Otherwise, the other differentiated regions without the TE variants have been already investigated with divergence scans based on SNPs. The total number of windows for F_{ST} calculation when considering all SNPs was 103 687, compared to 12 943 (12.5%), when considering only windows with TE variants.

We suppose the loci with variation in TEs do not have to necessarily overlap with differentiated loci based on non-TE variants. The autotetraploids arose relatively recently (Arnold et al., 2015) and unlikely have reached the mutational balance equilibrium (Monnahan et al., 2019). We expect the relaxed purifying selection mainly on low frequency TE variants, which are not shared with diploids as shown by Baduel et al., 2019. As a consequence of the recent origin of autotetraploids, there might have not been enough time for SNP mutations to occur within such regions.

8- From the methods section or the manuscript itself, it was unclear to me how the authors assessed the significance of the difference in MCL for different models of convergent evolution (e.g., Based on figure 2e, selection on de novo mutation has only a slightly higher likelihood than selection on standing variation – how did the authors determine the significance of the difference in MCL?). Were the parallel models only compared to the neutral model or also compared to each other? Could the authors also clarify why they choose to position the selected site at 8 equally distributed location across the candidate genes? Wouldn't it make sense to set these positions based on LD decay?

We estimated the maximum composite likelihood (MCL) over the parameters for each model and compared these MCL estimates among three parallel selection models. Then, we further used simulated data (from *A. arenosa*; Bohutínská, Vlček, et al., 2021) under the neutral model to find out which difference in MCLs between the parallel selection and neutral model is significantly higher than expected under neutrality. Briefly, they generated a distribution of differences between

selection model MCLs and the neutral MCL by analyzing neutral datasets, simulated with ms (Hudson, 2002), that had similar numbers of segregating sites and demographic history as our real data. We considered the MCL difference between a parallel and neutral model significant if it was higher than the maximum of the distribution of the differences from the simulated data (i.e. 21, a conservative estimate, Bohutínská, Vlček, et al., 2021). We clarified model selection in Methods I. 759-766:

“To choose the best fitting scenario for each candidate, we firstly estimated the maximum composite log-likelihood (MCL) over the parameters for each of the three parallel selection scenarios and a neutral scenario. We selected the parallel selection model with highest MCL and considered it significantly non-neutral only if the MCL difference between this parallel model and the neutral model was higher than the maximum of the distribution of the differences from the simulated neutral data in A. arenosa inferred in ref. Bohutínská, Vlček, et al., 2021 (i.e. MCL difference > 21, a conservative estimate).”

A similar approach was used by the original study introducing this approach (Lee and Coop, 2017) as well as additional empirical studies applying DMC (Van Etten et al., 2020; Wang et al., 2020). The reason for such an approach is that the use of a composite likelihood does not permit using standard asymptotic properties of likelihood estimators to construct confidence intervals or help with model choice (e.g., AIC).

We chose to place selected sites at eight locations at equal distance (default value recommended by authors <https://github.com/kristinmlee/dmc>) from each other along the particular gene. Such a density (one site per ~ 500 bp on average, as the mean length of serpentine adaptation candidate is ~ 4000 bp) is well within the range of the LD decay of 150-800 bp, that was estimated in A. arenosa (Bohutínská, Vlček, et al., 2021).

We clarified that also in the Methods I. 753-758:

“We chose to place selected sites at eight locations at equal distance (default value recommended by authors <https://github.com/kristinmlee/dmc>) from each other along the particular gene. Such a density (one site per ~ 500 bp on average, as the mean length of serpentine adaptation candidate is ~ 4000 bp) is in fact well-within the range of the LD decay of 150-800 bp, that was estimated in A. arenosa (Bohutínská, Vlček, et al., 2021).”

9-Line263: “Assuming linkage between TE variant and surrounding SNPs” -could this be formally tested?

We added to the text the explicit value of ± 100 bp (missing in the previous version of the text) I. 297-300. Such value is well below the estimated lower bound for the LD decay.

“Assuming linkage between each TE variant and surrounding SNPs (in the proximity of ± 100 bp), we applied a similar differentiation outlier window-based workflow as specified above and identified 92–115 TE-associated candidate genes per S-N contrasts (Dataset S9).”

10-Line 311 ‘from the three serpentine populations’. Could the authors please clarify why they focused on 3 populations?

We did additional Sanger sequencing only for S3, S4, S5 populations, because in S1 and S2 populations we did not observe any variation in *TPC1* locus from short-read resequencing data.

11-Line 372: “which have been shown to be important for stabilising Asn627 in A.thaliana”. Please provide the corresponding reference.

Corrected, we added the missing references l. 384-388:

“In the tertiary structure, residue 630 sits adjacent to the Asn residue (Asn627 in A. arenosa), which forms the pore’s constriction point and has been shown to control ion selectivity in A. thaliana (Guo et al., 2016; Guo et al., 2017; Kintzer and Stroud, 2016; Kintzer et al., 2018). In A. thaliana this Asn627 residue, when substituted by site-directed mutagenesis to the human homolog state can cause Na+ non-selective A. thaliana TPC1 to adopt the Na+ selectivity of human TPC1 (Guo et al., 2017).”

12-Line 442: “6” should be removed or replaced

Corrected

13-Line 603: “significantin”, should be “ significant in”

Corrected

14- Line 773: Shouldn’t “vcf” be “VCF”

Corrected

Reviewer #4 (Remarks to the Author):

This is an excellent study looking into the genes in the tetraploid A. arenosa that are leading to adaptations to serpentine soils, typically rich in Mg and Ni compared to non serpentine sites used for comparisons. The authors look for signatures of selection on serpentine sites and predict that most comes from existing allelic variation which is selected independently with the colonisation of serpentine soils. This is an important finding for those interesting in polyploidy species establishment and for those interested in understanding how species become tolerant to serpentine soils.

Thank you for the encouraging comments!

The work is very tightly written, at times too tightly written, so indefinite article are sometimes missing for my taste, but more critically, explanations of exactly what the

figures are showing is frequently minimal. No better example of this are the descriptions to Figure 2. This needs unpacking, and the legend and text appropriately expanded. The numbers on top of the Fig 2a bars means what, should they sum to 207? What do the bar colours mean (the legend being out of focus)? I think for 2e there is no reference at all about what we are to read from it.

Good point; thank you. We accordingly expanded the Fig. 2 legend I. 271-275:

“Intersection of candidates from each population pair (S1-N1 to S5-N5) demonstrating more genes repeatedly found as candidates across two, three and four population pairs than expected by chance alone (all intersections were significant at $p < 0.01$ (highlighted by asterisks), Fisher’s exact test, Dataset S4d); note: the colour intensity of the bars represents the p value significance of the intersections.”

We note that the numbers of observed overlaps should not sum to 207 as multiple candidates are shared among multiple population pairs. Indeed, the numbers on the top of the Fig. 2a (and also the similar graphs in the Supplementary figures) are not needed as the numbers of overlaps can be inferred from the y-axis, so we removed them.

Regarding Fig. 2e we slightly modified the legend I. 282-284:

“Two examples of parallel candidate loci, illustrating SNP divergence and maximum composite log-likelihood (MCL) estimation of the source of the selected alleles in these particular loci inferred in DMC.”

This unpacking is true also of Figure 3. Do we need ‘d’ at all? The yellow sticks are not discussed in the text I don’t think , we are told there are three colours, blue, marine and purple, I can resolve only two ‘blue-based’ colours. The text says “the Tyr side chain can adopt one of two broad conformations, either sticking into the channel, where it occludes the opening, or sticking away from the channel and directly into surrounding residues.....”. When I compare the expanded boxes I struggle to see what is the point being made. The feature they highlight needs to be arrowed or something...so the reader can compare S and N (the expanded bits are not always being identical to the unexpanded source does not help).

It is good practice in structural biology to show a large structure from multiple angles as it is invaluable in assisting the reader to situate important features within the broader structure. For example, in ‘d’ relative to ‘e’, ‘f’ and ‘g’ one can see that the residue of interest is close to the vacuole lumen side of the protein, distant from the cytosol-side gating control. Although we appreciate that one may need to already have familiarity with *TPC1* (or to have shown special interest in the *TPC1* structural papers cited), this extra perspective is none the less essential to allow the critical appraisal of our results by e.g. structural biologists and those who work on *TPC1*. Please see Fig. 3 legend I. 406-407:

The 'yellow sticks' are mentioned in the figure legend: "*The adjacent residue, 627 (631 in A. thaliana), which has an experimentally demonstrated key role in selectivity control, is yellow and drawn as sticks.*"

And further discussed in Results I. 386-388:

"In A. thaliana this Asn627 residue, when substituted by site-directed mutagenesis to the human homolog state can cause Na⁺ non-selective A. thaliana TPC1 to adopt the Na⁺ selectivity of human TPC1 (Guo et al., 2017)."

The point being made on I. 419-420 is that the two different conformations of the Tyr residue are both likely to be highly disruptive to the pore (particularly the functionally important residue Asn627). The feature being indicated in the figure is the difference between the conformations of the two bright-red residues. We have clarified this point further Results I. 415-417:

"Both of the conformations seen, shown in heterodimer form (Fig 3g), are highly likely to disrupt the stability of Asn627, thereby modifying the selectivity of the channel."

We also changed the purple color in Fig. 3d-g to lighter one to make the difference more visible.

There is also a necessity to better integrate the literature elsewhere, including from the authors themselves (Konečná et al., 2020), to compare these findings with what has been seen and reviewed before with regard to other serpentine systems. This includes previous findings (including in A. arenosa) using QTLs that have identified TPC1 and other genes for Ca, K and Ni homeostasis.

Thank you for this. We appreciate the suggestion to integrate the serpentine-related findings to other studies, but we also note that the main point of the current study is to determine the sources of adaptive variation in an autotetraploid; while we do point to previous studies showing certain candidate serpentine-adaptive alleles, we prefer to not further split the focus of this paper away from the main point, as it is already quite complex and long.

We have already mentioned Arnold et al., 2016 in the Introduction I. 97-99:

"As a proof-of-concept, selective ion uptake phenotypes and a polygenic basis for serpentine adaptation have been suggested from a single A. arenosa serpentine population (Arnold et al, 2016)."

Having asked for expansion of the text in all these areas, I remain confident that the work is overall excellent and a substantial contribution to our understanding of polyploid evolution and serpentine adaptations.

Thank you again for your very encouraging and helpful comments.

Additional points

- **Fig. 1. (A) plant picture is very poor and not needed, ideally the map of Europe should be expanded. The complementary figure in Supplementary materials needs to be coordinate with towns, rivers or something similar.**

Yes, we agree. We expanded the map in Fig. 1a and modified Fig. S1 according to the suggestions.

- **The text says “only the bioavailable concentration of Mg, Ni, Co, and Ca/Mg ratio consistently differentiated both soil types”. This is not the case for Mg and it should be removed.**

The concentration of Mg was indeed significantly differentiated between S and N soils, as tested by ANOVA and indicated in Methods I. 673-678:

“We tested the association of allele frequencies at each SNP for each individual with associated soil concentration of the key elements differentiating serpentine and non-serpentine soils: Ca/Mg ratio, and bioavailable soil concentrations of Co, Mg, and Ni. Only those elements were significant in one-way ANOVAs (Bonferroni corrected) testing differences in elemental soil concentration between S and N population, taking population pair as a random variable.”

- **The reference list appears twice.**

Corrected

- **There is a spurious ⁶**

Corrected

References

- Arnold, B. J., B. Lahner, J. M. DaCosta, C. M. Weisman, J. D. Hollister, D. E. Salt, K. Bomblies, and L. Yant. 2016. Borrowed alleles and convergence in serpentine adaptation. *Proceedings of the National Academy of Sciences* 113: 8320–8325.
- Arnold, B., S. T. Kim, and K. Bomblies. 2015. Single geographic origin of a widespread autotetraploid arabidopsis arenosa lineage followed by interploidy admixture. *Molecular Biology and Evolution* 32: 1382–1395.
- Baduel, P., B. Hunter, S. Yeola, and K. Bomblies. 2018. Genetic basis and evolution of rapid cycling in railway populations of tetraploid *Arabidopsis arenosa*. *PLoS Genetics* 14: 1–26.
- Baduel, P., L. Quadrana, B. Hunter, K. Bomblies, and V. Colot. 2019. Relaxed purifying selection in autopolyploids drives transposable element over-accumulation which provides variants for local adaptation. *Nature Communications* 10.
- Bohutínská, M., M. Alston, P. Monnahan, T. Mandáková, and S. Bray. 2021. Novelty and convergence in adaptation to whole genome duplication. *Molecular Biology and Evolution*: 1–37.
- Bohutínská, M., V. Handrick, L. Yant, R. Schmickl, F. Kolář, K. Bomblies, and P. Paajanen. 2021. De Novo Mutation and Rapid Protein (Co-)evolution during Meiotic Adaptation in *Arabidopsis arenosa*. *Molecular Biology and Evolution*: 1–15.
- Bohutínská, M., J. Vlček, S. Yair, B. Leanen, V. Konečná, M. Fracassetti, T. Slotte, and F. Kolář. 2021. Genomic

- basis of parallel adaptation varies with divergence in *Arabidopsis* and its relatives. *Proceedings of the National Academy of Sciences*: doi:10.1073/pnas.2022713118. Pre-print available at: 2020.03.24.005397.
- Boyko, A. R., S. H. Williamson, A. R. Indap, J. D. Degenhardt, R. D. Hernandez, K. E. Lohmueller, M. D. Adams, et al. 2008. Assessing the evolutionary impact of amino acid mutations in the human genome. *PLoS Genetics* 4.
- Van Etten, M., K. M. Lee, S. M. Chang, and R. S. Baucom. 2020. Parallel and nonparallel genomic responses contribute to herbicide resistance in *Ipomoea purpurea*, a common agricultural weed.
- Hollister, J. D., B. J. Arnold, E. Svedin, K. S. Xue, B. P. Dilkes, and K. Bomblies. 2012. Genetic Adaptation Associated with Genome-Doubling in Autotetraploid *Arabidopsis arenosa*. *PLoS Genetics* 8.
- Hudson, R. R. 2002. Generating samples under a Wright-Fisher neutral model of genetic variation. *Bioinformatics* 18: 337–338.
- James, M. E., M. J. Wilkinson, H. L. North, J. Engelstädter, and D. Ortiz-Barrientos. 2020. A framework to quantify phenotypic and genotypic parallel evolution. *bioRxiv*.
- Ji, Y., X. Li, T. Ji, J. Tang, L. Qiu, J. Hu, J. Dong, et al. 2020. Gene reuse facilitates rapid radiation and independent adaptation to diverse habitats in the Asian honeybee. *Science Advances* 6.
- Keightley, P. D., and B. C. Jackson. 2018. Inferring the Probability of the Derived vs. the Ancestral Allelic State at a Polymorphic Site. *Genetics* 209: 897–906.
- Lai, Y. T., C. K. L. Yeung, K. E. Omland, E. L. Pang, Y. Hao, B. Y. Liao, H. F. Cao, et al. 2019. Standing genetic variation as the predominant source for adaptation of a songbird. *Proceedings of the National Academy of Sciences of the United States of America* 116: 2152–2157.
- Lawrie, D. S., P. W. Messer, R. Hershberg, and D. A. Petrov. 2013. Strong Purifying Selection at Synonymous Sites in *D. melanogaster*. *PLoS Genetics* 9: 33–40.
- Lee, K. M., and G. Coop. 2017. Distinguishing among modes of convergent adaptation using population genomic data. *Genetics* 207: 1591–1619.
- Marburger, S., P. Monnahan, P. J. Seear, S. H. Martin, J. Koch, P. Pajanen, M. Bohutínská, et al. 2019. Interspecific introgression mediates adaptation to whole genome duplication. *Nature Communications* 10: 1–11.
- Meier, J. I., V. C. Sousa, D. A. Marques, O. M. Selz, C. E. Wagner, L. Excoffier, and O. Seehausen. 2017. Demographic modelling with whole-genome data reveals parallel origin of similar *Pundamilia* cichlid species after hybridization. *Molecular Ecology* 26: 123–141.
- Monnahan, P., and Y. Brandvain. 2020. The effect of autopolyploidy on population genetic signals of hard sweeps. *Biology Letters* 16.
- Monnahan, P., F. Kolář, P. Baduel, C. Sailer, J. Koch, R. Horvath, B. Laenen, et al. 2019. Pervasive population genomic consequences of genome duplication in *Arabidopsis arenosa*. *Nature ecology & evolution* 3: 457.
- Oziolor, E. M., N. M. Reid, S. Yair, K. M. Lee, S. Guberman VerPloeg, P. C. Bruns, J. R. Shaw, et al. 2019. Adaptive introgression enables evolutionary rescue from extreme environmental pollution. *Science* 364: 455–457.
- Pfeifer, S. P., S. Laurent, V. C. Sousa, C. R. Linnen, M. Foll, L. Excoffier, H. E. Hoekstra, and J. D. Jensen. 2018. The evolutionary history of Nebraska deer mice: Local adaptation in the face of strong gene flow. *Molecular Biology and Evolution* 35: 792–806.
- Preite, V., C. Sailer, L. Syllwasschy, S. Bray, U. Kraemer, and L. Yant. 2019. Convergent evolution in *Arabidopsis halleri* and *Arabidopsis arenosa* on calamine metalliferous soils. *Philosophical Transactions of the Royal Society B* 374.
- Schönswetter, P., I. Stehlik, R. Holderegger, and A. Tribsch. 2005. Molecular evidence for glacial refugia of mountain plants in the European Alps. *Molecular Ecology* 14: 3547–3555.
- Turner, T. L., E. C. Bourne, E. J. Von Wettberg, T. T. Hu, and S. V. Nuzhdin. 2010. Population resequencing reveals local adaptation of *Arabidopsis lyrata* to serpentine soils. *Nature Genetics* 42: 260–263.
- Wang, L., E. B. Josephs, K. M. Lee, L. M. Roberts, R. Rellán-Alvarez, J. Ross-Ibarra, and M. B. Hufford. 2020. Molecular Parallelism Underlies Convergent Highland Adaptation of Maize Landraces. *bioRxiv*: 2020.07.31.227629.
- Yant, L., J. D. Hollister, K. M. Wright, B. J. Arnold, J. D. Higgins, F. C. H. Franklin, and K. Bomblies. 2013. Meiotic adaptation to genome duplication in *Arabidopsis arenosa*. *Current biology* 23: 2151–2156.

REVIEWER COMMENTS

Reviewer #1 (Remarks to the Author):

The revision improved a lot, and I am satisfied with the present manuscript.

Reviewer #2 (Remarks to the Author):

I have now reviewed the revised manuscript in light of the authors' responses to all reviewers. In my opinion the authors have greatly improved the paper with their extensive and thoughtful revisions. I am now very happy to recommend that this paper be accepted for publication.

John Willis

Reviewer #3 (Remarks to the Author):

I would like to thank the authors for their careful answers to my comments. Most of my concerns have been addressed, and my interrogations have been clarified. Despite the impossibility to compare the present data with a diploid lineage, this work constitutes a very nice study of repeated adaptation. It confirms that footprints of selection at similar genomic positions during independent adaptations in autotetraploid mostly occur because of the repeated recruitment of standing variants. It also goes behind previous findings by presenting compelling evidence suggesting that adaptation by de novo mutations can indeed also occur in autotetraploid and by highlighting TPC1 as a mutational hotspot for serpentine soil adaptation. This work is, therefore especially interesting in the context of plants adaptation to serpentine soils.

I am still nevertheless unsure about the claim that adaptation is 'dominated' by the repeated recruitment of shared alleles. The proportion of shared differentiated regions seems low compared to the number of populations-specific genomic footprints, which suggests that serpentine adaptation is dominated by lineage-specific genetic changes.

Regarding the significance of the different 'source of adaptation' models, wouldn't it be possible to simulate data for other models than the neutral model (e.g., independent sweep) and determine if the difference in MCL of the observed data is higher than the maximum of the distribution of the difference from the simulated data? As I understand your approach test for departure from neutrality but do not per se compare the significance of different selection models. Could it be possible to use a similar parametric-bootstrapping approach to compare selection models? Would this change your interpretations?

Reviewer #4 (Remarks to the Author):

I remain of the view that the paper is excellent and interesting.

I said before, 'There is also a necessity to better integrate the literature elsewhere, including from the authors themselves (Konečná et al., 2020), to compare these findings with what has been seen and reviewed before with regard to other serpentine systems. This includes previous findings (including in *A. arenosa*) using QTLs that have identified TPC1 and other genes for Ca, K and Ni homeostasis'.

The authors have elected not to follow that advice.

The authors had previously, and still do state on line 266-269 of their manuscript 'Furthermore, when we compared our serpentine adaptation candidates (n=61) to candidate loci for parallel serpentine

adaptation in *A. lyrata* (n=62) from a previous study we found only two loci in common (significant overlap; $p < 0.007$), KUP9 and TPC1, further supporting important roles of these two ion transporters in repeated adaptation to serpentine soil.”

So why leave it at that? Why not report comparisons with other papers, including their own, and to the same species as that reported here, to add support to their assertion that there are ‘repeated sweeps’, and ‘significant parallelism’.

I am interested by referee 2’s point that ‘the overwhelming pattern is of unique adaptations in each serpentine population’. The authors agree to that point, but I really do not see that sentiment coming through in this manuscript. I also think that a comparison with the wider literature may contribute to this point too.

There is no reference in results to Fig 2e.

REVIEWER COMMENTS

Reviewer #1 (Remarks to the Author):

The revision improved a lot, and I am satisfied with the present manuscript.

Reviewer #2 (Remarks to the Author):

I have now reviewed the revised manuscript in light of the authors' responses to all reviewers. In my opinion the authors have greatly improved the paper with their extensive and thoughtful revisions. I am now very happy to recommend that this paper be accepted for publication.

John Willis

Reviewer #3 (Remarks to the Author):

I would like to thank the authors for their careful answers to my comments. Most of my concerns have been addressed, and my interrogations have been clarified. Despite the impossibility to compare the present data with a diploid lineage, this work constitutes a very nice study of repeated adaptation. It confirms that footprints of selection at similar genomic positions during independent adaptations in autotetraploid mostly occur because of the repeated recruitment of standing variants. It also goes behind previous findings by presenting compelling evidence suggesting that adaptation by de novo mutations can indeed also occurs in autotetraploid and by highlighting TPC1 as a mutational hotspot for serpentine soil adaptation. This work is, therefore especially interesting in the context of plants adaptation to serpentine soils.

I am still nevertheless unsure about the claim that adaptation is 'dominated' by the repeated recruitment of shared alleles. The proportion of shared differentiated regions seems low compared to the number of populations-specific genomic footprints, which suggests that serpentine adaptation is dominated by lineage-specific genetic changes.

Agreed. Thank you for this comment, this was our poor wording. Indeed, our study informs that *parallel* adaptation specifically (not serpentine adaptation most generally considered), displays widespread repeated recruitment. There have been misleading statements throughout which we now corrected (e.g. l. 1, 38, 112-114, 511-513). Specifically:

1) The misleading title was a result of our previous revision when we replaced "parallel" by "serpentine" without noticing such an important change in the meaning. To clarify our main result regards parallel not all serpentine candidates, we suggest returning to the original title (while keeping "serpentine" focus in the Abstract)

“Parallel adaptation in autopolyploid Arabidopsis arenosa is dominated by repeated recruitment of shared alleles”

2) We clarified this in the research questions (l. 112-114)

“(1) Does gene-level parallel adaptation in autotetraploid A. arenosa dominantly reflect repeated sampling from the large pool of shared variation that is expected to be maintained in autopolyploids?”

3) We also modified the statement in conclusion accordingly, to clarify we are referring to the parallel loci not all adaptive loci (l. 511-513):

“Footprints of selection at similar genomic positions mostly occur because of the repeated recruitment from a large pool of pre-existing variation, yet exceptionally also from recurrent de novo mutations.”

Regarding the significance of the different ‘source of adaptation’ models, wouldn’t it be possible to simulate data for other models than the neutral model (e.g., independent sweep) and determine if the difference in MCL of the observed data is higher than the maximum of the distribution of the difference from the simulated data? As I understand your approach test for departure from neutrality but do not per se compare the significance of different selection models. Could it be possible to use a similar parametric-bootstrapping approach to compare selection models? Would this change your interpretations?

We apologize that we had not commented on this particular question in previous revisions. In fact, the MCL simulations of neutral data for parallel selection models are unfortunately not possible in the genome-wide context. The reason is that unlike in neutral models where a general genome-wide background is used, these MCL differences for the non-neutral scenarios needed to be estimated case-by-case (per each locus and each pair of parallel population contrasts), for each already with particular parameter values (selection coefficients, migration rates and standing time varied among loci). Thus, such an approach would lead to non-uniform MCL thresholds applied, making an unified comparison among different loci genome-wide impossible. Because an overall comparison was our primary aim, we kept the standard way of comparing the parallel models in order to provide a straightforward genome-wide overview. Such an approach has been so far used in all other empirical implementations comparing multiple loci that we are aware of (Oziolor et al., 2019, Van Etten et al., 2020, Wang et al., 2020).

To clarify that we also compared MCL among the parallel models, not only parallel vs. neutral one, we changed the following statement in the Methods (l. 777-781):

“We selected among the parallel selection models by choosing the model with the highest MCL, following the approach of Lee and Coop 2017. Further, we considered the case significantly non-neutral only if the MCL difference between the selected parallel model and the

corresponding neutral model was higher than the maximum of the distribution of the differences from the simulated neutral data in A. arenosa ... ”

Admitting that such an approach does not allow comparison of the magnitude of MCL difference between the non-neutral parallel models, we do not comment on the significance of such difference in the ms. On the bright side, taking the *TPC1* locus as an example, the location of the proposed selected site, for which the *de novo* model had the highest MCL, indeed matches the region with the likely functional non-synonymous mutation at residue 630 (I.368-371), providing additional support for the *de novo* compared to the standing variation parallel model.

Reviewer #4 (Remarks to the Author):

I remain of the view that the paper is excellent and interesting.

I said before, ‘There is also a necessity to better integrate the literature elsewhere, including from the authors themselves (Konečná et al., 2020), to compare these findings with what has been seen and reviewed before with regard to other serpentine systems. This includes previous findings (including in *A. arenosa*) using QTLs that have identified *TPC1* and other genes for Ca, K and Ni homeostasis’.

The authors have elected not to follow that advice.

The authors had previously, and still do state on line 266-269 of their manuscript ‘Furthermore, when we compared our serpentine adaptation candidates (n=61) to candidate loci for parallel serpentine adaptation in *A. lyrata* (n=62) from a previous study we found only two loci in common (significant overlap; $p < 0.007$), *KUP9* and *TPC1*, further supporting important roles of these two ion transporters in repeated adaptation to serpentine soil.’

So why leave it at that? Why not report comparisons with other papers, including their own, and to the same species as that reported here, to add support to their assertion that there are ‘repeated sweeps’, and ‘significant parallelism’.

Added. We expanded our interpretation of our findings in the light of other studies (please see below and on I.271-282 and the new Table S5). We agree such a discussion may further be useful for direct comparisons with other plant species adapted to serpentines. But at the same time, the overall scarcity of the sufficiently investigated systems unfortunately does not permit generalisations (genome-wide only *Arabidopsis lyrata* and *Mimulus guttatus*).

*“In addition, when overlapping the candidate genes detected at least in one of our five population pairs with serpentine *A. lyrata* study we revealed additional convergent loci involved in ion homeostasis, calcium, nickel, and potassium transmembrane transport (Table S5), suggesting existence of ‘hotspot’ regions in *Arabidopsis* genome in response to serpentine stress. An additional candidate gene (*FPN2* = *IREG2*) investigated in *Alyssum* (*Brassicaceae*;*

Sobczyk et al., 2017) has been found to be shared between three population pairs. Finally, when comparing to the only genomically investigated serpentine system outside Brassicaceae (*Mimulus*, Phrymaceae; Selby 2014), there was only limited overlap in two loci with one our population pair, despite overall similar functions were enriched, suggesting parallel adaptation though similar pathways in spite of limits to gene-level parallelism in very divergent (~140myr) species.”

Table S5 Gene coding loci exhibiting excessive differentiation (1% outlier F_{ST}) between particular S-N population pair in *A. arenosa* which have been also identified as candidates for serpentine adaptation in other available plant studies.

Species	Family	A. arenosa S1-N1	A. arenosa S2-N2	A. arenosa S3-N3	A. arenosa S4-N4	A. arenosa S5-N5
Arabidopsis lyrata ¹	Brassicaceae	AT3G15730, AT4G19440, AT5G09650, AT1G31120, AT1G69730, AT5G03570, AT4G19960	AT1G72560, AT4G03560, AT4G19960	AT2G46140, AT3G01310, AT4G32640, AT4G12430, AT1G51310, AT1G31120, AT4G03560, AT4G19960	AT2G24070, AT3G46520, AT1G51310, AT4G03560, AT5G03570	AT4G34450, AT1G69730, AT5G03570
Alyssum serpyllifolium ²	Brassicaceae	AT5G03570			AT5G03570	AT5G03570
Mimulus guttatus ³	Phrymaceae				AT4G19880, AT4G19670	

¹(Turner et al., 2010), N of candidates = 62

²(Sobczyk et al., 2017) N of candidates = 2

³(Selby, 2014), N of candidates = 10

Please note that the additional genes overlapping *A. lyrata* and *A. arenosa* that were presented in our review (Konečná et al., 2020) reflect a comparison with the only one *A. arenosa* population that has been investigated and published by that time (Arnold et al., 2016; NB this

study used serpentine population that was also directly involved in our study, S3). We note that there has been no QTL study on serpentine adaptation conducted in *A. arenosa*. In our review (Konečná et al., 2020) we refer to QTL study in *Mimulus guttatus* (Selby 2014; Selby and Willis 2018). The *TPC1* locus has previously been identified based on the selection scans only (Turner et al., 2010; Arnold et al., 2016) but not using a QTL study.

I am interested by referee 2's point that 'the overwhelming pattern is of unique adaptations in each serpentine population'. The authors agree to that point, but I really do not see that sentiment coming through in this manuscript. I also think that a comparison with the wider literature may contribute to this point too.

We agree strongly. This is an excellent point. Indeed, we further checked the text and removed any vague quantification statements on "substantial" parallelism. We further refer to a "significant" parallelism in terms of non-random overlap in the number of genes throughout the text (e.g. l. 108, 230-235, l. 270, l. 286, l. 318).

"The level of parallelism was greater than expected by chance for all pairs of S-N contrasts (Fisher's exact test; Fig 2a, Dataset S4d) and we hereafter refer to "significant parallelism". Such a fraction of parallel gene candidates (0.02-0.04 out of all candidates from that particular population pair) is in line with other naturally adapting systems of comparable divergence (Takuno et al., 2015; Lai et al., 2019; Preite et al., 2019; Bohutínská et al., 2021).

In addition, we explain that the observed proportion of parallel candidates among all candidates detected within each pairwise combinations (0.02-0.04) is well within the range of other studies investigating genome-wide parallelism among populations of comparable divergence and provide references for those recent studies (mean proportion of parallel candidates: Takuno et al., 2015 - 0.018; Lai et al., 2019 - 0.052; Preite et al., 2019 - 0.019 and 0.026; Bohutínská et al., 2021 - 0.045).

Finally, we also consistently clarified throughout the text the dominant fraction refers to *parallel* adaptive candidates, e.g. l. 1, l. 115, l. 119, l. 351, l. 458.

There is no reference in results to Fig 2e.

Corrected

REVIEWERS' COMMENTS

[Editor: Reviewer #3 states in Remark to Editor section that (s)he is satisfied with the revision.]

Reviewer #4 (Remarks to the Author):

A very nice paper, line 282 needs a little editorial work to improve the syntax

Responses to Reviewers

[Editor: Reviewer #3 states in Remark to Editor section that (s)he is satisfied with the revision.]

Reviewer #4 (Remarks to the Author):

A very nice paper, line 282 needs a little editorial work to improve the syntax

Done, syntax is changed

Reporting summary

Updated reporting summary, following the Editorial comments, is uploaded

Author checklist

A revised author checklist describing our response to our editorial requests is uploaded.